# FMRP regulates mRNAs encoding distinct functions in the cell body and dendrites of CA1 pyramidal neurons

Caryn R Hale[1]*, Kirsty Sawicka[1], Kevin Mora[1], John J Fak[1], Jin Joo Kang[1], Paula Cutrim[1], Katarzyna Cialowicz[2], Thomas S Carroll[3], Robert B Darnell[1,4]*

[1]Laboratory of Molecular Neuro-Oncology, Rockefeller University, New York, United States; [2]Bio-Imaging Resource Center, The Rockefeller University, New York, United States; [3]Bioinformatics Resource Center, The Rockefeller University, New York, United States; [4]Howard Hughes Medical Institute, Rockefeller University, New York, United States

**Abstract** Neurons rely on translation of synaptic mRNAs in order to generate activity-dependent changes in plasticity. Here, we develop a strategy combining compartment-specific crosslinking immunoprecipitation (CLIP) and translating ribosome affinity purification (TRAP) in conditionally tagged mice to precisely define the ribosome-bound dendritic transcriptome of CA1 pyramidal neurons. We identify CA1 dendritic transcripts with differentially localized mRNA isoforms generated by alternative polyadenylation and alternative splicing, including many that have altered protein-coding capacity. Among dendritic mRNAs, FMRP targets were found to be overrepresented. Cell-type-specific FMRP-CLIP and TRAP in microdissected CA1 neuropil revealed 383 dendritic FMRP targets and suggests that FMRP differentially regulates functionally distinct modules in CA1 dendrites and cell bodies. FMRP regulates ~15–20% of mRNAs encoding synaptic functions and 10% of chromatin modulators, in the dendrite and cell body, respectively. In the absence of FMRP, dendritic FMRP targets had increased ribosome association, consistent with a function for FMRP in synaptic translational repression. Conversely, downregulation of FMRP targets involved in chromatin regulation in cell bodies suggests a role for FMRP in stabilizing mRNAs containing stalled ribosomes in this compartment. Together, the data support a model in which FMRP regulates the translation and expression of synaptic and nuclear proteins within different compartments of a single neuronal cell type.

*For correspondence:
chale@rockefeller.edu (CRH);
darnelr@rockefeller.edu (RBD)

**Competing interest:** The authors declare that no competing interests exist.

## Editor's evaluation

The authors performed transcriptomic analyses from compartment-specific, micro-dissected hippocampal region tissue from transgenic mice. One feature that distinguishes this work from previous FMRP studies is the use of conditional knock-in tags (GFP or HA) and tissue-specific expression of the Cre recombinase to target a population of pyramidal neurons in the CA1 region. The strengths of the paper are the rich datasets and innovative integration of methods that will provide a valuable technical resource for the field.

## Introduction

A key feature in the molecular biology of learning and memory is protein synthesis-dependent synaptic plasticity, which involves translation of localized mRNAs in response to synaptic activity. Local translation has been demonstrated in neuronal dendrites and axons (reviewed in *Glock et al., 2017*; *Lin and*

**eLife digest** The brain has over 100 billion neurons that together form vast networks to relay electrical signals. A neuron receives electrical signals from other neurons via branch-like structures known as dendrites. The signals then travel into the cell body of the neuron. If their sum reaches a threshold, they fire a new signal through a single outgoing projection known as the axon, which is connected to the dendrites of other neurons.

A single neuron has thousands of dendrites that each receive inputs from different axons, and it is thought that the strengthening and weakening of these dendritic connections enables us to learn and store memories.

Dendrites are filled with molecules known as messenger ribonucleic acids (mRNAs) that act as templates to make proteins. Axonal signals reaching the dendrites can trigger these mRNAs to make new proteins that strengthen or weaken the connections between the two neurons, which is believed to be necessary for generating long-term memories.

A protein called FMRP is found in both the cell body and dendrites and is able to bind to and regulate the ability of mRNAs to make proteins. A loss of the gene encoding FMRP is the most common cause of inherited intellectual disability and autism in humans, but it remains unclear precisely what role this protein plays in learning and memory.

Hale et al. used genetic and bioinformatics approaches to specifically study mRNAs in the dendrites and the cell body of a specific type of neuron involved in memory in mice. The experiments revealed that FMRP played different roles in the dendrites and cell body. In the dendrites, FMRP interacted with mRNAs encoding proteins that can change how the neuron responds to a signal from a neighboring neuron and may alter how strong the connections between the neurons are. On the other hand, FMRP in the cell body modulated the activities of mRNAs encoding proteins that in turn regulate the activities of genes.

These findings change the way we think about how memory may work by suggesting that groups of mRNAs encoding proteins with certain activities are found in distinct parts of a single neuron. These observations offer new ways to approach intellectual disabilities and autism spectrum disorder.

*Holt, 2007*; *Rangaraju et al., 2017*) and allows for rapid and precise changes in the local proteome near active synapses. In dendrites, a brief burst of local translation has been shown to be necessary and sufficient for induction of the late phase of long-term potentiation (L-LTP, occurring hours to days after potentiation) (*Frey et al., 1988*; *Kang and Schuman, 1996*; *Kang et al., 1997*) and long-term depression (LTD) (*Huber et al., 2000*), and inhibiting protein synthesis blocks long-term memory formation (*Frey et al., 1988*; *Sutton and Schuman, 2006*).

Activity-dependent local translation depends on both the availability of specific mRNAs and the sensitivity with which their translation can be initiated upon local signaling events. Both rely on interactions between mRNAs, a host of RNA-binding proteins, and ribosomes. mRNAs are thought to be transported in a translationally repressed state into the neuronal processes via transport granules containing RNA-binding proteins such as the Fragile-X mental retardation protein (FMRP), CPEB1, ZBP-1, and STAU1/2 (*Hüttelmaier et al., 2005*; *Krichevsky and Kosik, 2001*; *Martin and Ephrussi, 2009*). Although dendritic targeting elements have been defined for a few mRNAs such as *Camk2α*, *Actb*, and *Map2* (*Andreassi and Riccio, 2009*), and a few RNA-binding proteins have been found to regulate individual mRNAs, such as the interaction between ZBP-1 and the *β-actin* mRNA (reviewed in *Biswas et al., 2019*), the functional relationship between the global dendritic transcriptome and individual RNA-binding proteins is still largely unknown. For at least some localized mRNA granules, signaling cascades initiated by synaptic activity lead to their dissolution and initiation of translation (*Dahm and Kiebler, 2005*), but the role of RNA regulatory factors in this process is incompletely understood. The integrated study of the dendritic transcriptome and the RNA-binding proteins responsible for regulation of local translation will provide critical insight into the mechanisms underlying protein synthesis-dependent synaptic plasticity.

FMRP, the RNA-binding protein whose activity is lost in Fragile X syndrome, represses translation (*Bassell and Warren, 2008*; *Costa-Mattioli et al., 2009*; *Darnell et al., 2011*; *Laggerbauer et al., 2001*) and is thought to be a key regulator of activity-dependent local translation in neurons

(*Banerjee et al., 2018*; *Bear et al., 2004*; *Huber et al., 2002*; *Lee et al., 2011*). Dendritic FMRP levels are increased upon neuronal activity, with evidence for local translation of the FMRP transcript itself (*Greenough et al., 2001*; *Weiler et al., 1997*) and kinesin-mediated movement of FMRP-containing mRNA transport granules from the neuronal cell body (*Dictenberg et al., 2008*). At the synapse, FMRP is proposed to be linked to local signal transduction, potentially through calcium-induced post-translational modification of the protein, which alters the FMRP granule and leads to translation of the mRNAs (*Lee et al., 2011*; *Narayanan et al., 2007*). FMRP knockout (KO) neurons show excess basal translation as well as an inability to produce activity-stimulated translation (*Ifrim et al., 2015*).

Direct FMRP targets have been identified in the whole mouse brain through CLIP studies (*Darnell et al., 2011*; *Korb et al., 2017*), indirectly through ribosome-binding studies (translating ribosome affinity purification [TRAP]; *Ceolin et al., 2017*; *Kumari and Gazy, 2019*) or through TRAP together with crosslinking immunoprecipitation (CLIP) (*Sawicka et al., 2019*). Some recent studies have explored FMRP targets specifically in the excitatory CA1 neurons of the mouse hippocampus (*Ceolin et al., 2017*; *Sawicka et al., 2019*). FMRP target genes overlap significantly with autism susceptibility genes and include genes involved in both synaptic function and transcriptional control in the nucleus (*Darnell, 2020*; *Darnell et al., 2011*; *Iossifov et al., 2012*; *Sawicka et al., 2019*), and loss of FMRP increases translation of chromatin modifiers such as BRD4 (*Korb et al., 2017*) and SETD2 (*Shah et al., 2020*). These and other observations have suggested a model in which FMRP regulates the stoichiometry of its targets in two ways: globally, by translational control of transcription regulators in the cell body, and locally, by enabling activity-dependent local translation of synaptic proteins in dendrites (*Darnell, 2020*), but it is still unclear the extent to which such regulation occurs simultaneously in a single neuron. Here, we probe this model by exploring subcellular compartment-specific patterns of FMRP binding and regulation.

We utilize compartment- and cell-type-specific profiling technologies to precisely define the transcriptome of mouse hippocampal CA1 pyramidal neurons. We use TRAP and conditionally tagged (cTag) mice that express tagged RNA-binding proteins in a single-cell type to study RNA regulation specifically in CA1 neurons combined with manual microdissection to isolate compartment-specific proteins and mRNAs. RNA profiling of subcellular CA1 compartments reveals that dendritic mRNAs are enriched for elongated 3′UTR isoforms and depleted for alternative splicing (AS) events driven by the neuronal splicing factor NOVA2, indicating a nuclear role in the generation of the localized transcriptome in CA1 neurons. Integrating compartment-specific cTag-FMRP-CLIP and TRAP defined FMRP CLIP scores in the dendrites and cell bodies of CA1 neurons and identified 383 FMRP-bound dendritic targets. This allowed us to distinguish FMRP targets according to their site of regulation within neurons, revealing enrichment of FMRP-regulated mRNAs encoding nuclear proteins in the CA1 cell bodies and mRNAs encoding synaptic proteins in the CA1 dendrites. Moreover, although mRNA localization is unaffected in FMRP KO mice, mRNAs encoding these synaptic proteins show altered localized ribosome association. Together, these findings support a model in which distinct patterns of both mRNA and FMRP subcellular localization enable FMRP to regulate the expression of different proteins within different compartments in a single neuronal cell type.

## Results

### Identification and characterization of the dendritic transcriptome in hippocampal pyramidal neurons in vivo

We developed a system that allows for parallel isolation of mRNAs and RNA-binding proteins that are enriched in the cell bodies or dendrites specifically in excitatory CA1 neurons in the hippocampus (*Figure 1A*). We created three CA1-specific protein-tagged mouse lines by crossing cTag mice with mice in which Cre recombinase expression is driven from the Camk2a promoter (*Tsien, 1998*). In these mice, Cre is expressed only in pyramidal neurons of the hippocampus. The cTag-PABP (*Hwang et al., 2017*) and cTag-FMRP mice allow for Cre-dependent expression of GFP tagged polyA-binding protein c1 (PABPC1), , or FMRP (*Sawicka et al., 2019*; *Van Driesche et al., 2019*), respectively. The RiboTag (*Sanz et al., 2009*) mouse allows for Cre-driven expression of HA tagged RPL22, a ribosomal subunit. Crossing these animals with Camk2a-Cre mice results in lines expressing tagged ribosomes, or in the case of cTag polyA-binding protein c1 (PABPC1 or FMRP, 'knock-in' tagged proteins

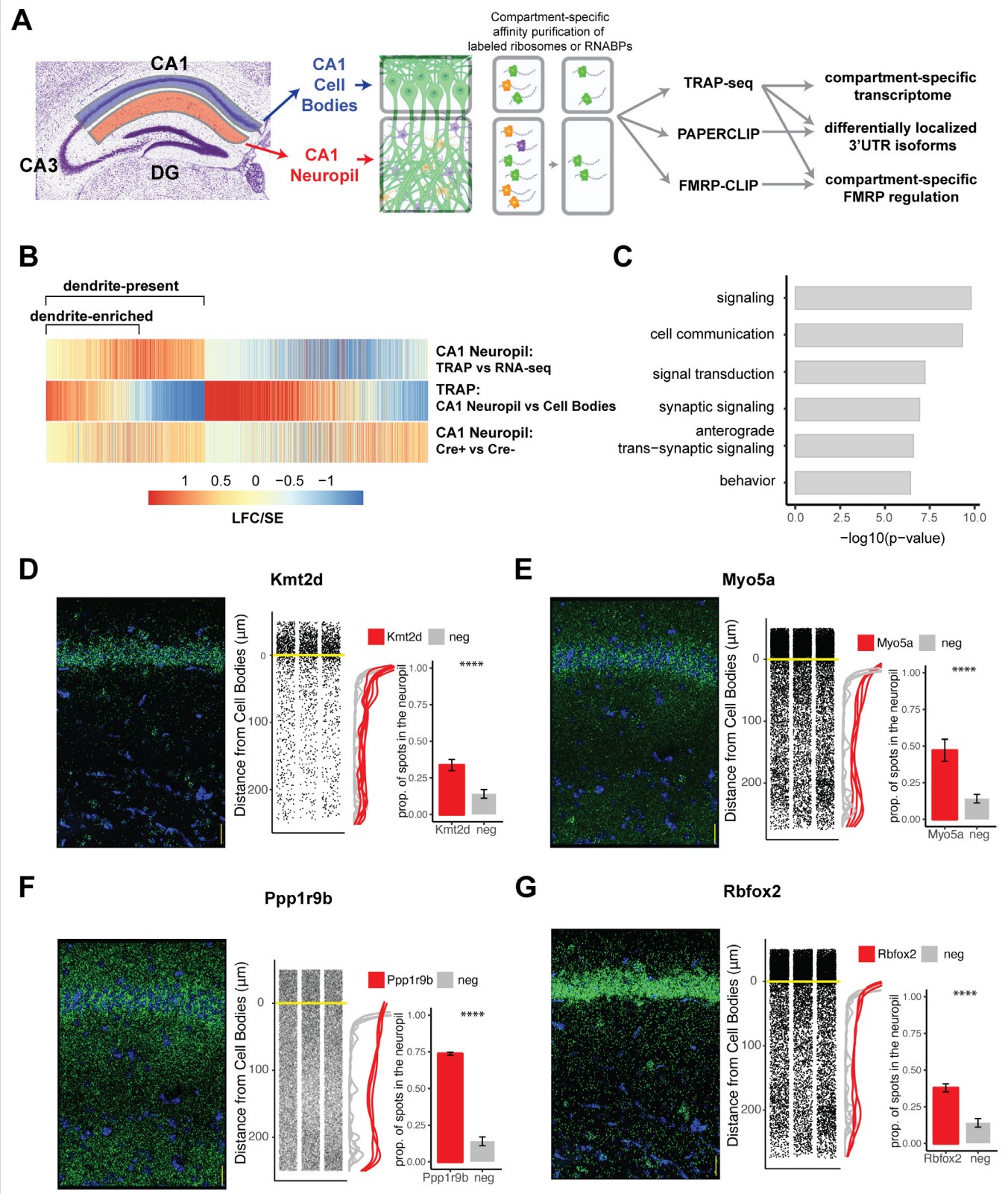

**Figure 1.** Combining cell-type-specific protein tagging and manual microdissection allows for precise definition of the CA1 dendritic transcriptome. (**A**) Experimental design. Hippocampal slices from Camk2a-Cre-expressing conditionally tagged mouse lines were subject to microdissection in order to separate the CA1 cell bodies and CA1 neuropil layers. These layers contained material from pyramidal neurons (in which proteins of interest contain an affinity tag, green) and contaminating cell types (other colors). Microdissected compartments were subject to affinity purification in order to obtain

*Figure 1 continued on next page*

*Figure 1 continued*

pyramidal neuron-specific ribosomes or affinity-tagged RNA-binding proteins and bound mRNAs. In order to obtain the dendritic ribosome-bound transcriptome, TRAP-seq was performed from tagged ribosomes in the CA1 neuropil compartment. Compartment-specific cTag-PAPERCLIP was performed in order to determine mRNAs with 3′UTR isoforms that undergo differential localization, and compartment-specific FMRP regulation was determined by cTag FMRP-CLIP of the microdissected compartments. (**B**) Identification of dendritic mRNAs. Differential gene expression analysis was performed on bulk RNA-seq and TRAP-seq from microdissected CA1 compartments. All CA1-expressed mRNAs as shown. Colors indicate the log2 fold change (LFC)/SE (standard error, stat) from DESeq analysis. mRNAs significantly enriched in CA1 neuropil TRAP over bulk RNA-seq of the CA1 neuropil were defined as 'dendrite-present.' mRNAs that were also significantly enriched in CA1 neuropil TRAP when compared to cell bodies TRAP were considered to be 'dendrite-enriched.' In addition, only mRNAs that were enriched in CA1 neuropil TRAP in Camk-Cre-expressing RiboTag mice when compared to RiboTag mice not expressing Cre were considered. (**C**) Localized mRNAs are highly enriched for genes involved in synaptic signaling and synapse organization. GO analysis was performed comparing dendrite-enriched mRNAs to all mRNAs expressed in CA1 neurons. (**D–G**) Validation of localized mRNAs. Fluorescence in situ hybridization (FISH) was performed using the RNAscope method using probes designed against the entire mRNA of the indicated gene. Left: representative FISH image of RNAscope on the CA1 region of coronal brain sections. mRNA spots are shown in green, and DAPI staining is shown in blue. Scale bars represent 30 μm. Middle: the distance between the mRNA punctae and the cell bodies was quantitated for three representative images. Density is plotted for all collected images (red) and compared to a negative control mRNA (*Snca*) that was identified as sequestered in the cell bodies (gray). Right: mRNAs more than 10 μm from the cell bodies were considered to be in the neuropil. The proportion of indicated mRNAs that were found in the neuropil is plotted. A cell body-sequestered mRNA (*Snca*) is used as a negative control (neg). Asterisks indicate results of the Wilcoxon rank-sum test (****p<0.00001).

The online version of this article includes the following figure supplement(s) for figure 1:

**Figure supplement 1.** CA1 compartment-specific TRAP-seq enriches for dendritic mRNAs derived specifically from Camk2a-expressing pyramidal neurons.

**Figure supplement 2.** Characteristics of CA1 dendritic mRNAs.

expressed from native genes. In the hippocampus, this expression is specific in the CA1 pyramidal neurons (*Figure 1—figure supplement 1A*; *Hwang et al., 2017*).

Microdissection of the CA1 neuropil compartments and immunoprecipitation (IP) allowed us to enrich dendritic tagged proteins that originated from the cell bodies (CBs) of the CA1 neurons. TRAP-seq (*Heiman et al., 2014*), in which mRNAs bound to affinity-tagged ribosomes are immunoprecipitated and sequenced, enriched for mRNAs bound to dendritic ribosomes (in Camk2a-Cre x RiboTag mice). cTag-PAPERCLIP (conditionally tagged poly(A)-binding protein-mediated mRNA 3′ end retrieval by crosslinking immunoprecipitation) (*Hwang et al., 2017*) allowed cell-type-specific CLIP of the polyA-binding protein, PABPC1, and subsequent sequencing of 3′UTR-polyA tail junctions in order to determine the precise end of the 3′UTR of an expressed mRNA. Here, we use cTag-PAPERCLIP on microdissected CA1 cell bodies and neuropil to describe compartment-specific 3′UTR usage. cTagFMRP-CLIP (in Camk2a-Cre × Fmr1-cTag mice) allowed us to identify compartment-specific FMRP-binding events, extending previous studies (*Darnell et al., 2011*; *Sawicka et al., 2019*).

Microdissected CA1 compartments from 8- to 10-week-old mice were subjected to bulk RNA-seq as a denominator for all transcripts in the neuropil, and TRAP as a denominator for all CA1 pyramidal neuron-specific, ribosome-bound dendritic transcripts. IP conditions were optimized to isolate relatively pure, intact, ribosome-bound mRNAs with minimal contamination by interneurons and glial cells found in the neuropil (*Figure 1—figure supplement 1B–E*). As a negative control, RiboTag animals not expressing the Cre recombinase were microdissected and subject to affinity purification and sequencing, and only mRNAs enriched over these controls were considered for downstream analyses. We identified two groups of dendritic, ribosome-bound mRNAs: dendrite-present (significantly enriched in CA1 neuropil TRAP-seq over CA1 neuropil bulk RNA-seq; 2058 mRNAs) and dendrite-enriched (dendrite-present and also significantly enriched in CA1 neuropil TRAP over cell bodies TRAP; 1211 mRNAs; *Figure 1B*, see *Supplementary file 1A and B* for a full list of mRNAs identified). 689 (34%) of the dendrite-present mRNAs were previously identified in bulk RNA-seq of the microdissected rat CA1 neuropil (*Cajigas et al., 2012*), and these RNAs were found to be significantly enriched in dendrites (CA1 neuropil TRAP vs. bulk RNA-seq; *Figure 1—figure supplement 2A and B*).

The identified dendrite-enriched mRNAs were significantly longer than the whole-cell transcriptome identified in CA1 pyramidal neurons (*Sawicka et al., 2019*), whether considering full-length transcripts, 5′UTR, 3′UTR, or coding sequence (CDS) portions (*Figure 1—figure supplement 2C*). Gene Ontology (GO) analysis of dendrite-enriched mRNAs showed strong enrichment for genes encoding proteins with important roles in the synapse such as synaptic signaling, anterograde synaptic signaling, and behavior (*Figure 1C*), consistent with prior analyses (*Cajigas et al., 2012*).

We used RNAscope fluorescence in situ hybridization (FISH) to validate the presence in dendrites of several mRNAs that had not been identified in previous studies including *Kmt2d* (a histone methyl-transferase), *Myo5a* (an actin motor protein involved in transporting cargo to dendrites), *Ppp1r9b* (a scaffolding protein component of protein phosphatase 1a important for dendritic spine morphology), and *Rbfox2* (a neuronal splicing factor) (*Figure 1D–G*). Interestingly, approximating the distance from the cell body for each detected mRNA spot revealed variable mRNA distribution patterns for different transcripts, suggesting multiple potential paths for mRNA localization. For example, roughly 35% of the transcripts encoding *Kmt2d* and *Rbfox2* were detected throughout in the neuropil, whereas ~74% of the transcripts encoding *Ppp1r9b* were abundant in the distal neuropil (*Figure 1F*). By comparison, less than 15% of mRNAs encoding alpha-synuclein, a neuronal gene whose protein product is involved in presynaptic transmission (*Snca*) and an mRNA identified as enriched in the CA1 cell body compartment, were found in the CA1 neuropil.

## Identification of mRNAs with 3′UTR isoforms preferentially localized to dendrites

Subcellular localization of cytoplasmic mRNAs is thought to be at least partially mediated by 3′UTR elements (*Andreassi and Riccio, 2009*; *Blichenberg et al., 2001*; *Mayford et al., 1996*; *Tushev et al., 2018*). However, analysis of 3′UTRs from RNA-seq data alone is complicated by mixed cell types, incomplete annotation, and difficulty in identifying internal polyA sites. To identify the expressed 3′UTRs in CA1 pyramidal neurons, we first used polyA sites determined by Camk2a-Cre-driven cTag-PAPERCLIP from whole hippocampus (*Hwang et al., 2017*) and microdissected CA1 compartments to define all 3′ ends. We next used splice junctions identified from TRAP to define 5′ end of each 3′UTR (*Figure 2—figure supplement 1*). This allowed us to identify the boundaries of potential 3′UTRs (*Figure 2A*, *Figure 2—figure supplement 2A and B*), and revealed 15,322 3′UTRs expressed in Camk2a-expressing pyramidal neurons, including 3700 genes that give rise to mRNAs with more than one 3′UTR isoform. Analyzing expression of these 3′UTRs in the compartment-specific TRAP data revealed 219 3′UTR isoforms that were differentially localized to CA1 dendrites (*Figure 2B*, *Supplementary file 1D*).

Analysis of these differentially localized 3′UTRs revealed transcripts generated by two types of alternative polyadenylation (APA), distinguished by their effect on the CDS of the resulting protein. APA events that do not affect the CDS of the resulting protein derive from transcripts with multiple polyadenylation sites in a single 3′UTR, resulting in isoforms with short (proximal) and long (distal) 3′UTRs (3′UTR-APA). APA events that truncate the CDS of the resulting protein utilize polyA sites in upstream regions, resulting in multiple (short and long) protein isoforms (UR-APA) (*Tian and Manley, 2017*). Of the 219 genes producing differentially localized 3′UTR isoforms in CA1 neurons, we found that 149 had no effect on the CDS, 48 resulted in altered CDS, and 22 generated both event types (*Figure 2B*, left panel). Among isoforms with unchanged CDS, distal 3′UTRs were significantly enriched in dendrites, consistent with a previous study of CA1 neuropil RNAs analyzed by 3′ end sequencing (*Tushev et al., 2018*). Conversely, proximal 3′UTRs were significantly enriched in the CA1 cell bodies (*Figure 2B*, middle and right panels). We used FISH to validate these types of differential localization events, including Calmodulin 1 (*Calm1*) (*Figure 2—figure supplement 2B–E*), previously described to harbor differentially localized 3′UTR isoforms (*Tushev et al., 2018*), F-box protein 31 (*Fbxo31*) (*Figure 2—figure supplement 2F–H*), an E3 ubiquitin ligase proposed to be involved in neuronal maintenance and dendritic outgrowth (*Vadhvani et al., 2013*), and vesicle-associated membrane protein B (*Vapb*) (*Figure 2—figure supplement 2I–K*), a membrane protein involved in vesicle trafficking.

Approximately 20% of the differential isoform localization events (48 out of 219) involved a polyadenylation event that led to an extension or truncation of the CDS (*Figure 2B*). For example, the gene for connector enhancer of kinase suppressor of Ras2 (*Cnksr2* or MAGUIN) produces mRNAs with two 3′UTRs isoforms: a short isoform that is highly sequestered in the cell bodies (less than 10% of transcripts were found in the CA1 neuropil by FISH) and a longer isoform of which at least 40% of transcripts were localized in the CA1 neuropil (*Figure 2C–E*). Analysis of the ankyrin repeat and sterile alpha motif domain containing 1B (*Anks1b*) gene revealed differential localization of an isoform generated from 5′ extension of the 3′UTR sequence, which was depleted in the CA1 cell bodies, and again validated by FISH (*Figure 2F–H*). Finally, two mRNAs produced from the ank-repeat domain containing protein 11 (*Ankrd11*) gene were identified, a full-length version that contains Ank repeats,

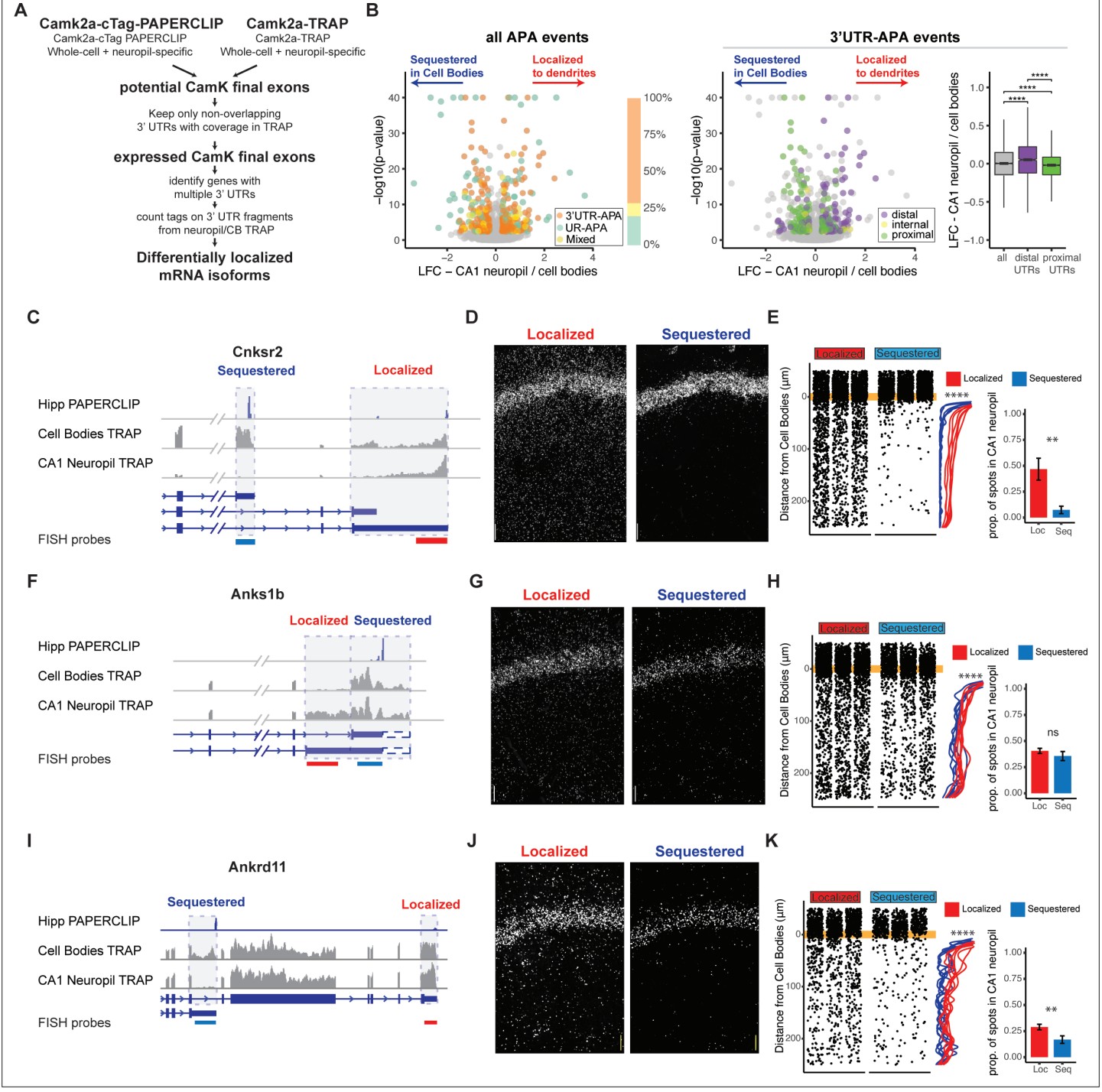

**Figure 2.** Combining cTag-PAPERCLIP and TRAP in order to identify genes with differentially localized 3'UTR isoforms. (**A**) Scheme for identification of expressed 3'UTR isoforms in CA1 neurons followed by analysis of differential localization. Boundaries of expressed 3'UTR isoforms in CA1 neurons were determined by combining polyA sites, determined by cTag-PAPERCLIP from both the whole hippocampus and microdissected CA1 compartments, with splice junctions from cell-type-specific TRAP experiments. These potential final exons were filtered for nonoverlapping 3'UTRs with complete coverage in TRAP. Compartment-specific expression of the resulting 3'UTRs was quantitated, and DEXSeq was used to determine 3'UTRs that were differentially localized in the dendrites of CA1 neurons. (**B**) Differential localization of 3'UTR isoforms. Left: volcano plot shows log2 fold change (LFC) (calculated by DEXSeq) vs. false discovery rate (FDR) for genes that produce significant differentially localized 3'UTR isoforms, colored by types of alternative polyadenylation (APA) events. 3'UTR-APA (orange) are 3'UTRs with multiple polyA sites; APA changes do not affect the coding sequence (CDS) of the resulting mRNAs. Genes that undergo upstream region APA, or UR-APA (green), utilize polyA sites within introns upstream of the 3'UTR, and result in mRNAs with truncated CDS. Genes that undergo both types of APA are shown in yellow. Proportion of significant events that fall into each of these

*Figure 2 continued on next page*

*Figure 2 continued*

groups is summarized in bar graph, with the same color scheme as the volcano plot. Middle: the 3'UTR-APA events shown in left are here colored according to their position in the gene, either proximal to the stop codon (green), internal (yellow), or distal (purple). Right: for all 3'UTR-APA events detected, distal 3'UTRs (purple) are significantly enriched in the CA1 neuropil (determined by the LFC of CA1 neuropil/cell bodies) when compared to proximal (green) 3'UTRs. p-Values for paired Wilcoxon rank-sum tests are indicated (****p<0.00001). (**C–K**) Validation of differential localization of 3'UTRs by fluorescence in situ hybridization (FISH). (**C**) Differential localization of *Cnksr2* 3'UTR isoforms. Distribution of cell bodies- and CA1 neuropil TRAP-seq reads for the 3' end of the *Cnksr2* mRNA. Camk2a-cTag-PAPERCLIP tags from the hippocampus are shown in blue. Coverage is normalized for read depth and scaled in order to best illustrate isoform expression. Predicted mRNA isoforms are indicated below, and the positions of the FISH probes are indicated (the sequestered probe is shown in blue, and the localized probe in red). (**D**) smFISH on the CA1 region using probes against localized (left) and sequestered (right) 3'UTR sequences. (**E**, left) Spots were counted in either the CB or CA1 neuropil region and distance traveled from the cell body was determined for each spot. Plots show location of spots in all quantitated replicates. Line plots show the density of the detected spots that were found in the CA1 neuropil in either the cell body-sequestered (blue) or neuropil-localized 3'UTR isoform (red) for the 300 nts proximal to the cell bodies. Asterisks indicate significance in Kolomogorov–Smirnov tests (****p<0.00001) between distribution of sequestered and localized 3'UTR isoforms. Right: overall quantitation of spots in the cell bodies (<10 µm from the cell body layer) and CA1 neuropil (>10 µm from the cell bodies) is shown in barplots. Results of Wilcoxon rank-sum tests are shown (**p<0.001). (**F–H**) Differential localization of *Anks1b* 3'UTR isoforms. See description for (**C–E**). Dashed box indicates a potential underutilized 3'UTR extension that is observed by TRAP, but represents only a minor fraction of PAPERCLIP reads. (**I–K**) Differential localization of *Ankrd11* 3'UTR isoforms.

The online version of this article includes the following figure supplement(s) for figure 2:

**Figure supplement 1.** CA1 compartment-specific cTag-PAPERCLIP.

**Figure supplement 2.** Differentially localized 3'UTR-APA events.

as well as the C-terminal transcriptional repression and activation domain, and a previously uncharacterized isoform derived from a polyadenylation site found in intron 8, which is able to produce a protein that contains only the Ank-repeat regions (see PAPERCLIP profile in *Figure 2I*). The truncated isoform was predominantly detected in the cell bodies of CA1 neurons by both TRAP and FISH, while the full-length isoform was detected in both the cell bodies and dendrites (*Figure 2I–K*). Together, these data demonstrate the utility of combining compartment- and cell-type transcriptomics and PAPERCLIP to define expressed 3'UTRs and reveal that dendritic transcripts with altered protein-coding capacity are generated by alternative processing of 3'UTRs.

## Identification of mRNAs with AS isoforms that are preferentially localized to the dendrites

We next sought to identify alternative spliced RNA isoforms that were differentially abundant in the dendrites of CA1 pyramidal neurons. After analysis with rMATS (*Shen et al., 2014*) and filtering, we identified 165 AS events in 143 genes that were differentially expressed between the two compartments (*Figure 3A*, *Supplementary file 1E*). Of these, 106 (64.2%) were skipped exons, 32 (19.4%) were alternative 3' splice sites, 14 (8.5%) were alternative 5' splice sites, and 13 (7.9%) were mutually exclusive exons (*Figure 3B*). These alternatively spliced transcripts encode proteins involved in synaptic functions such as action potential, receptor localization, and synaptic signaling, as well as mRNA splicing (*Figure 3C*).

To determine splicing factors that may be responsible for these differentially localized AS events, we used existing datasets (see *Supplementary file 1F*) of splicing changes previously found to be mediated by neuronal AS factors. Of these, MBNL1/2 (using data from *Weyn-Vanhentenryck et al., 2018*) and NOVA2 (using data from *Saito et al., 2016*) were found to regulate the largest number of these events (37 for MBNL1/2 and 36 for NOVA2, *Figure 3D*). Interestingly, we found that CA1 neuropil/cell body splicing changes were positively correlated with splicing changes in NOVA2 KO animals (from analysis of *Nova2*-null vs. WT data, Pearson coefficient = 0.498, p-value=9.87e-08), which indicates that NOVA2 drives splicing changes that result in mRNAs that are preferentially sequestered in CA1 cell bodies (*Figure 3E*, left panel). This effect was specific for NOVA2 as MBNL1/2-dependent splicing changes did not show such a correlation with the localized splicing changes (Pearson coefficient = –0.00438, p-value=0.9698, *Figure 3E*, right panel).

Among transcripts that exemplify differential exon usage in dendritic transcripts were *Rapgef4/Epac2* and neuronatin (*Nnat*). *Rapgef4* (*Epac*), the gene encoding a cAMP-activated guanine exchange factor for RAP1 and RAP2 involved in LTP in the hippocampus, expresses two isoforms in the brain, a full-length isoform (*Epac2A1*), and one that is lacking exon 7 (*Epac2A2*) (*Hoivik et al.,*

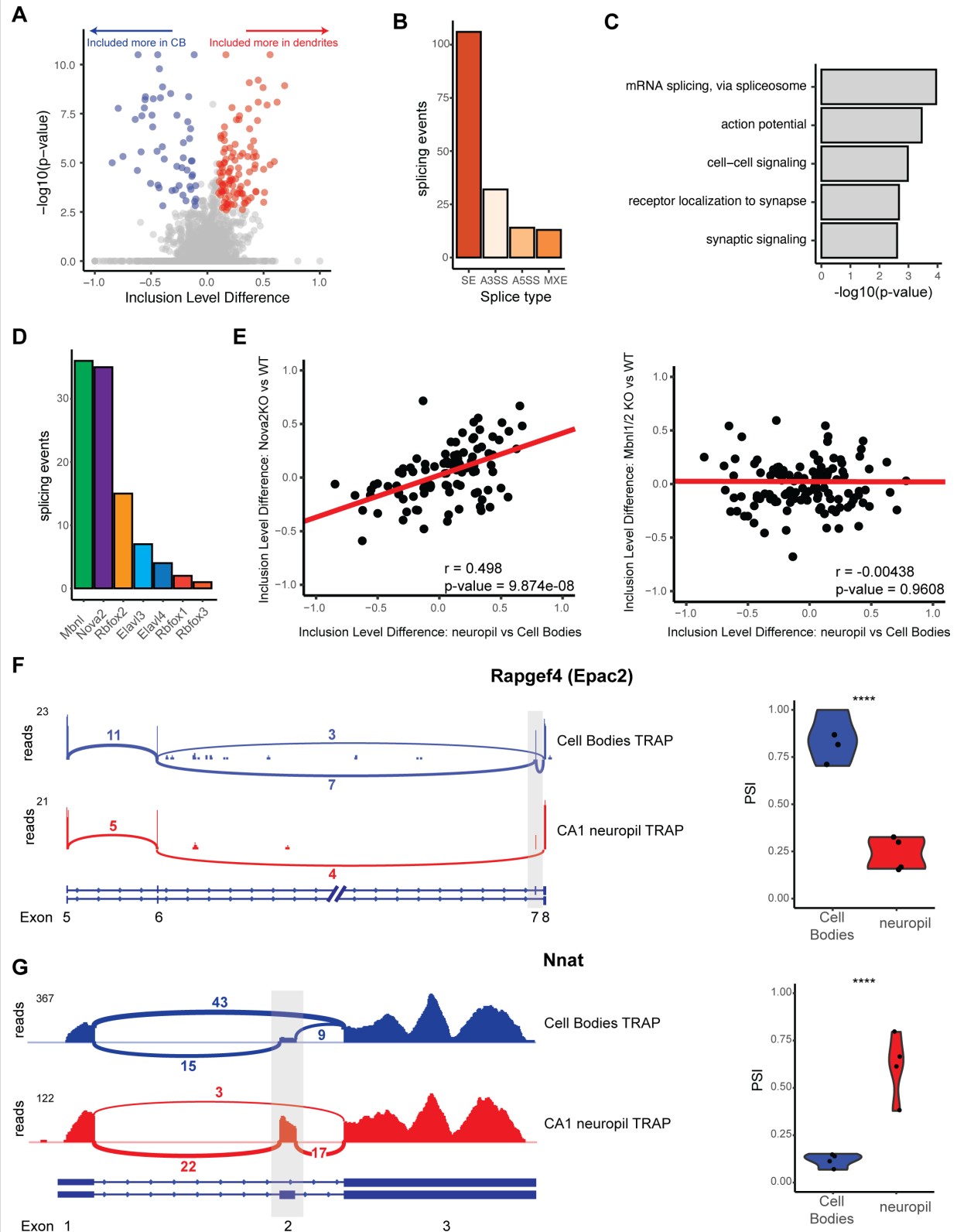

**Figure 3.** Differential localization of mRNAs with alternative splice events. (**A**) Analysis of cell bodies and CA1 neuropil TRAP by rMATS reveals differentially abundant alternative splice events. Volcano plot shows the inclusion level difference vs. the -log10(p-value) for each detected splice event. Significant events (false discovery rate [FDR] < 0.05, |dPSI| > 0.1) are colored either red (included more in the CA1 dendrites) or blue (included more in cell bodies). (**B**) Types of splicing events identified as differentially localized. (**C**) GO analysis reveals enriched functional terms for mRNAs with

*Figure 3 continued on next page*

*Figure 3 continued*

differentially localized alternative splice events. All mRNAs expressed in CA1 neurons were used as a background. (**D**) Neuronal RNA-binding proteins that are responsible for differentially localized alternative splicing (AS) events. AS analysis was performed on RNA-binding protein KO vs. WT RNA-seq data (see *Supplementary file 1F* for sources of data). Splicing events that were shown to be differentially localized (seen in **A**) and also changed in the absence of the RNA-binding protein are plotted. (**E**) NOVA2 neuronal splicing factor generates cell body-restricted mRNA transcripts. Inclusion-level differences in CA1 neuropil vs. cell bodies-TRAP-seq are compared with the inclusion-level differences in NOVA2-/- vs. WT RNA-seq data (left) and MBNL1/2-/- RNA-seq (right). Red line indicates a fitted linear model of the data. Results of the Pearson correlation test are shown. (**F**) Differential localization of spliced *Rapgef4* (Epac2) mRNAs. Representative Sashimi plots (left) are shown for cell bodies (blue) and CA1 neuropil (red) TRAP-seq. Exon numbers are indicated. Coverage indicates aligned reads. Numbers of detected splice junctions are shown. Violin plots (right) show the percent spliced in (PSI) values for the alternative splice event shown in the Sashimi plot. Each dot represents a single TRAP-seq replicate. Asterisks indicate significance (FDR) of the splicing change, as determined by rMATS (****FDR < 0.00001). (**G**) Differential localization of spliced *Nnat* mRNAs (see **F** for description).

*2013*). Of the *Rapgef4* transcripts detected in the CA1 dendrites, only 25% included exon 7, whereas in the cell bodies, 75% of the *Rapgef4* contained exon 7 (*Figure 3F*), indicating preferential localization of the transcripts without exon 7 to the CA1 dendrites. *Nnat*, a maternally imprinted gene whose protein is important for regulation of intracellular calcium levels, is expressed as either an α- and β-isoform in which exon 2 is included or skipped, respectively. We found that *Nnat* transcripts lacking exon 2 are predominantly sequestered in the cell bodies, with only ~12.5% of cell body transcripts containing exon 2. Conversely, the majority of localized *Nnat* transcripts (50–75%) contain exon 2, indicating preferential localization of the exon 2 containing *Nnat* transcripts (*Figure 3G*). These observations underscore the role of AS in generation of localized transcript isoforms. More generally, these data demonstrate that dendritic transcripts with altered protein-coding capacity are generated by both APA and AS.

## CA1 FMRP targets are overrepresented in the dendritic transcriptome

FMRP is thought to be a master regulator of local translation (*Ronesi and Huber, 2008*), leading us to examine the relationship between the FMRP targets previously defined in CA1 neurons (*Sawicka et al., 2019*) and those that we found to be present in the dendritic ribosome-bound transcriptome (*Figure 1*). We observed significant overrepresentation of FMRP targets in dendrite-present mRNAs, and even more so in dendrite-enriched mRNAs (*Figure 4A*). Of 1211 dendrite-enriched mRNAs, about 35% (413 mRNAs) were FMRP targets compared to 28.5% of dendrite-present mRNAs and 11.6% of all CA1-expressed mRNAs (*Figure 4B*).

We next compared the relative abundance (as compared to the cell bodies) of three groups of mRNAs: all CA1 FMRP targets, and dendrite-enriched mRNAs that either are or are not CA1 FMRP targets. Dendrite-enriched FMRP targets were significantly more abundant in dendrites than nontargets (*Figure 4C*). Further characterization of these dendrite-enriched mRNAs revealed that they were generally longer than all CA1-expressed mRNAs (*Figure 1—figure supplement 2*), but that FMRP-bound dendritic mRNAs were significantly longer than the nontargets (p-value=9.13e-46, *Figure 4D*). These observations were consistent with prior observations that FMRP preferentially binds long mRNAs (*Darnell et al., 2011*; *Sawicka et al., 2019*), and taken together, suggest that FMRP binds the majority of long, dendritic mRNAs.

Examination of the functional differences between dendrite-enriched FMRP targets and nontargets revealed an enrichment in dendritic FMRP targets for proteins involved in synaptic signaling, behavior, regulation of trans-synaptic signaling, and GTPase-mediated signal transduction (*Figure 4E*). These data indicate that FMRP is a key regulator of local translation in the dendrite of mRNAs encoding proteins involved in important synaptic functions.

Previous work on mRNA localization in FMRP KO cells in vitro has suggested a role for interactions between G-quadruplexes in the 3'UTRs of FMRP target mRNAs and the RGG-domain of the FMRP protein (*Goering et al., 2020*). We examined dendrite-enriched FMRP targets for enrichment of potential G-quadruplexes. Importantly, we found that all dendrite-enriched mRNAs are highly G- and C-rich (*Figure 4—figure supplement 1B–D*), so we analyzed differences in G-quadruplex containing transcripts between dendrite-enriched FMRP targets and dendrite-enriched non-FMRP targets (*Figure 4—figure supplement 1A*). We searched for experimentally defined G-quadruplexes (*Guo and Bartel, 2016*; *Figure 4—figure supplement 1E*) and also predicted G-quadruplex motifs (as defined in *Goering et al., 2020*) in the 3'UTRs of dendrite-enriched FMRP targets and FMRP

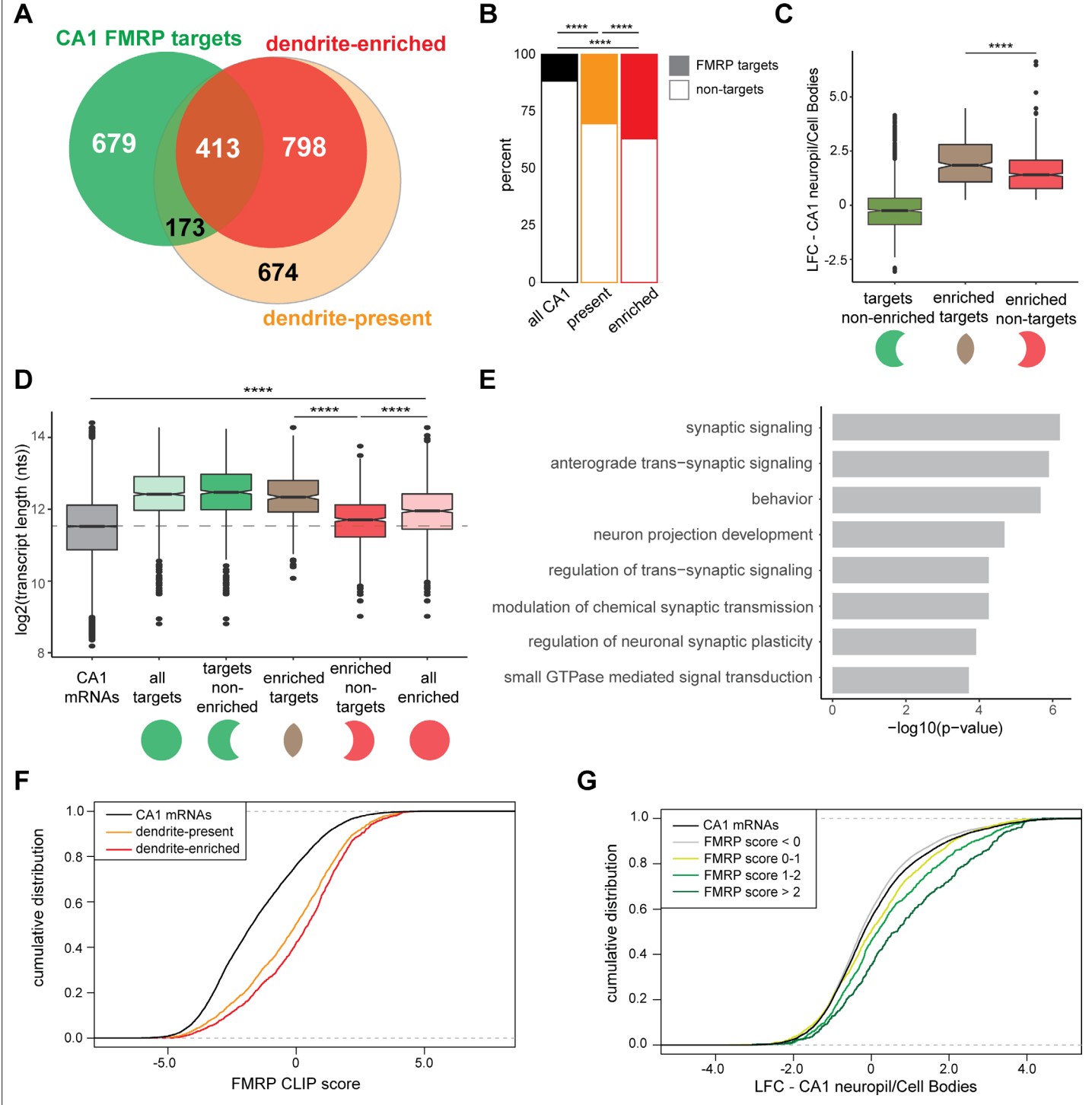

**Figure 4.** CA1 FMRP targets are overrepresented in the dendritic ribosome-associated transcriptome. (**A**) Overlap of CA1 FMRP targets and the dendritic transcriptome defined by CA1 neuropil TRAP. CA1 FMRP targets are defined as those with FMRP CLIP-scores > 1 in hippocampal CamK-cTag-FMRP (*Sawicka et al., 2019*). Dendrite-enriched and dendrite-present mRNAs are defined in *Figure 1*. (**B**) CA1 FMRP targets are more overrepresented in the dendrite-enriched mRNAs than in the dendrite-present mRNAs. Chi-squared analysis was performed to determine the enrichment of CA1 FMRP targets among dendrite-present mRNAs (p-value=1.42e-175) and dendrite-enriched mRNAs (p-value=1.70e-170) in comparison to the fraction of FMRP targets that are expressed in CA1 neurons (as defined by CA1-specific TRAP). (**C**) CA1 FMRP targets are highly abundant in dendrites. Dendrite-enriched mRNAs were subdivided into CA1 FMRP targets (enriched targets) and nontargets (enriched nontargets), and the dendritic abundance (defined by log2 fold change [LFC] CA1 neuropil/cell bodies in DESeq2 analysis) was compared for each group. Dendrite-enriched mRNAs that are also CA1 FMRP targets are significantly more abundant in dendrites than dendritic non-FMRP targets. Wilcoxon rank-sum test was used to determine

*Figure 4 continued on next page*

Figure 4 continued

significance (****p<0.00001). (**D**) CA1 FMRP targets in the dendritic transcriptome are significantly longer than non-FMRP targets. mRNA transcript lengths (in log2(nts)) for all CA1-expressed genes and the subsets defined in (**A**) were compared. For each gene expressed in the CA1 transcriptome, the length of the most highly expressed mRNA was considered. Wilcoxon rank-sum test was used to determine significance. Dashed line indicates the mean transcript length for all CA1 mRNAs. (**E**) CA1 FMRP targets in the dendritic transcriptome encode proteins involved in synaptic signaling and synaptic plasticity. GO analysis was performed by comparing the dendrite-enriched CA1 FMRP targets (enriched targets) with all dendrite-enriched mRNAs. (**F**) CA1 FMRP targets in the dendritic transcriptome have large CA1 FMRP CLIP scores. CA1 FMRP CLIP scores for all CA1 genes were determined previously for whole-cell FMRP cTag CLIP and CA1-specific TRAP. Cumulative distribution function (CDF) plots compare the CA1 FMRP CLIP scores (*Sawicka et al., 2019*) for all CA1 genes (black) and those defined as either dendrite-present (orange) or dendrite-enriched (red). (**G**) Relative abundance in dendrites of CA1 FMRP targets found in the dendritic transcriptome correlates with FMRP binding. Relative abundance in dendrites (LFC/ standard error [SE] CA1 neuropil TRAP vs. cell bodies TRAP) was compared by CDF plots for all CA1 genes (black) and subsets with CA1 FMRP-CLIP scores less than 0, 0–1, 1–2, or over 2.

The online version of this article includes the following figure supplement(s) for figure 4:

**Figure supplement 1.** Sequence characteristics of FMRP targets found in the dendrites.

nontargets (*Figure 4—figure supplement 1F*). We found no evidence for significant enrichment of G-quadruplexes in dendrite-enriched FMRP targets.

FMRP 'CLIP scores' were previously developed as a metric to define FMRP-bound transcripts with greater amount of FMRP binding relative to other transcripts of similar abundance in CA1 neurons (*Sawicka et al., 2019*). Dendrite-enriched mRNAs had significantly higher FMRP CLIP scores and hence greater FMRP binding than the dendrite-present group (p-value=2.646e-05, *Figure 4F*). Additionally, FMRP CLIP scores positively correlated with relative abundance in dendrites: when CA1 mRNAs were grouped according to the magnitude of their CA1 FMRP CLIP scores, those with increasingly higher scores were increasingly abundant in dendrites (*Figure 4G*). Taken together, these results suggest that FMRP binds mRNAs that are more abundant in dendrites than in cell bodies. Moreover, the magnitude of CA1 FMRP CLIP scores are predictive of the relative dendritic abundance of its targets (*Figure 4G*).

## FMRP selectively binds dendritic mRNA isoforms

We examined whether differential transcript isoforms were specifically bound by FMRP in hippocampal CA1 neurons. For example, the *Ankrd11* transcript undergoes APA to express a short and long isoform, and only the long isoform is abundant on dendritic ribosomes (*Figure 2I–K*). Interestingly, CA1 FMRP-CLIP tags were detected on the long, dendritic isoform, but only sparsely on the short isoform (*Figure 5A*, gray dashed boxes). To look at this phenomenon on a transcriptome-wide scale, we isolated exon junction reads in whole hippocampus CA1 FMRP-cTag-CLIP data. While the length of CLIP tags (20–100 nts) results in a low number of junction reads, we were able to confidently identify FMRP-CLIP tags covering 17 differentially abundant alternative splice events. For example, FMRP binding was largely absent on a shorter, CB-enriched isoform of the *Cnksr2* transcript, while robust binding was evident on the longer, dendritic 3'UTR (*Figure 5B*, gray dashed boxes). Of the 12 exon junction reads that originated from exon 20 of the *Cnksr2* transcript, 10 were derived from the long isoform, suggesting that approximately 80% of the FMRP-bound *Cnksr2* transcripts derived from the longer, dendritic isoform. This was especially striking since the shorter isoform was the predominant isoform in CA1 pyramidal neurons (~80% of exon junction reads in cell body TRAP belonged to the short isoform), indicating a high degree of selectivity of FMRP binding to this dendritic isoform (*Figure 5B*). Globally, we compared the percent spliced in (PSI) values for the 17 detected alternative splice events detected in FMRP-CLIP with those in the CA1 cell body and neuropil TRAP data. This revealed that splicing events identified in FMRP-bound mRNAs show stronger correlation with PSI values determined in CA1 neuropil TRAP relative to cell body TRAP (*Figure 5C*). Taken together, these results indicate that FMRP preferentially binds to specific processed transcripts that are fated for dendritic localization.

## Identification of dendritic FMRP targets

In order to identify direct FMRP-bound mRNA targets in CA1 dendrites, we crossed FMRP cTag mice with Camk2a-Cre mice, tagging FMRP with GFP specifically in the CA1 pyramidal neurons (*Figure 1A*). Hippocampal slices from cTag mice were crosslinked, microdissected into cell body and

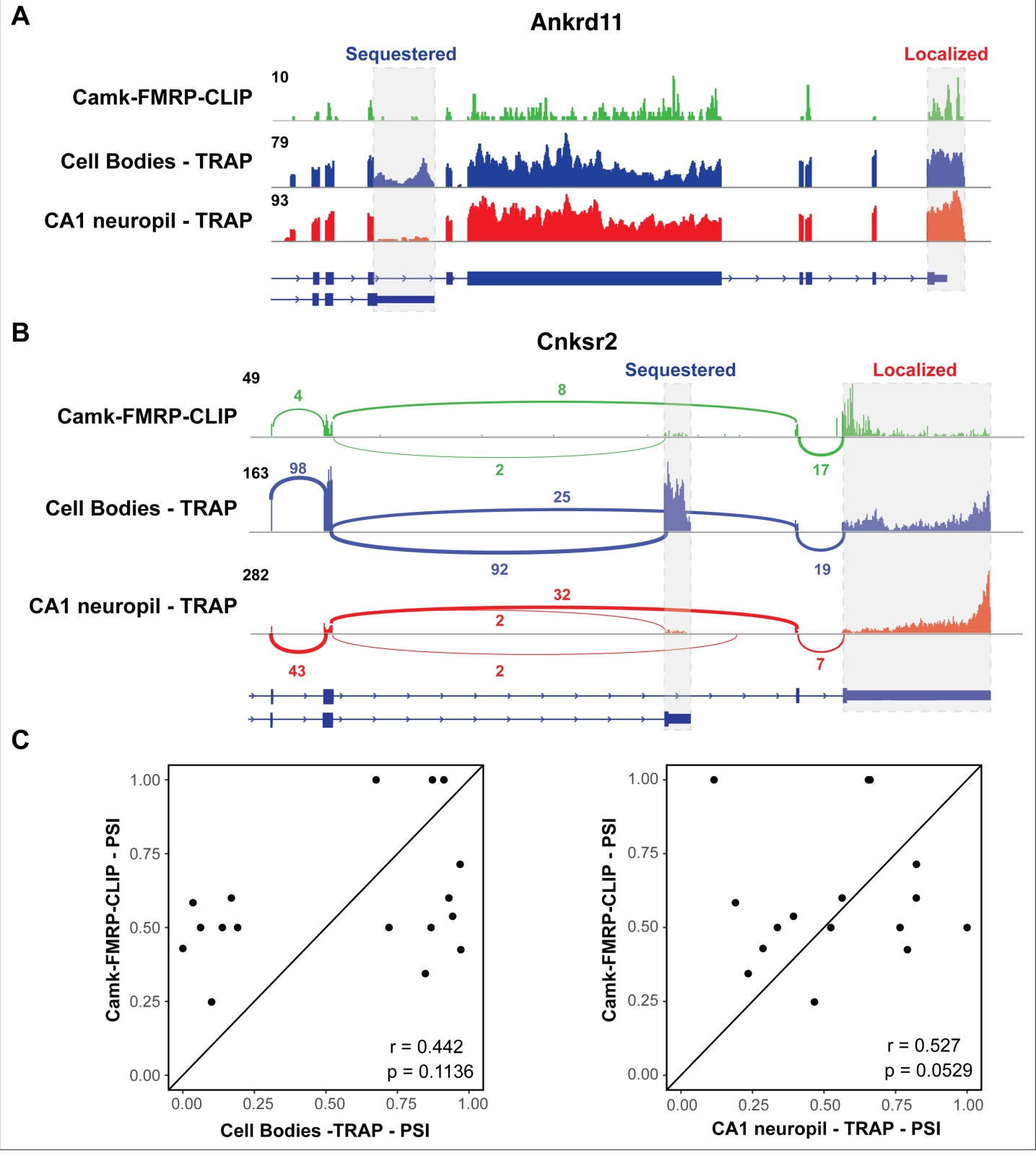

**Figure 5.** FMRP specifically binds localized mRNA isoforms. (**A**) FMRP preferentially binds long, localized *Ankrd11* mRNAs. The *Ankrd11* gene encodes mRNAs with two potential 3′UTRs (gray boxes). CA1 FMRP-CLIP tags (from hippocampal Camk-cTag FMRP CLIP reported previously; *Sawicka et al., 2019*) are shown in green, and representative coverage from CA1 cell bodies (blue) and CA1 neuropil-(red) TRAP is shown. (**B**) Splice junctions in FMRP-CLIP derived from *Cnksr2* mRNA isoforms. *Cnksr2* expresses two mRNA isoforms, indicated by gray boxes. Sashimi plots illustrate coverage and

*Figure 5 continued on next page*

*Figure 5 continued*

junction-spanning reads from CA1 FMRP-CLIP in green (tags are aggregated from three replicates). Sashimi plots are also shown for cell bodies TRAP (blue) and CA1 neuropil TRAP (red). (**C**) Splicing isoforms discovered in FMRP-CLIP tags resemble those found in the localized transcriptome. Percent spliced in (PSI) values derived from splice junction reads in CA1 FMRP-cTag-CLIP tags were compared to PSI values from the same events in cell bodies-TRAP (left) and CA1 neuropil TRAP (right). Results of Pearson correlation tests are shown.

neuropil regions, and subjected to FMRP-CLIP using antibodies against GFP. This allowed purification of FMRP-bound RNA specifically in the CA1 cell bodies or dendrites. Across five biological replicates, we obtained 746,827 FMRP CA1-specific CLIP tags from the cell bodies and 80,749 tags from CA1 dendrites. Overall, we observed a similar distribution of FMRP CLIP tags across the CDS in these mRNAs and in the two compartments (*Figure 6—figure supplement 1*), consistent with prior CLIP analysis and the general observation that FMRP binds CDS to arrest ribosomal elongation (*Darnell et al., 2011*).

Combining compartment-specific TRAP and FMRP-CLIP experiments allowed us to determine compartment-specific FMRP CLIP scores for the CA1 cell bodies and dendrites (*Figure 6A*, *Figure 6—figure supplement 2*, *Supplementary file 1G*). From this, we identified 383 'dendritic FMRP targets,' defined as mRNAs that are reproducibly bound by FMRP in CA1 dendrites (*Supplementary file 1H*). Of these dendritic FMRP targets, 60.8% (233) were mRNAs defined in *Figure 1* as dendrite-enriched (*Figure 6B*) and 76.5% (293) were dendrite-present (*Figure 6—figure supplement 1*). Dendritic FMRP targets show greater relative abundance in ribosome-bound mRNAs (TRAP) when compared to all CA1 FMRP targets (*Figure 6C*). Additionally, when comparing the FMRP-CLIP scores identified previously by whole hippocampus CA1 FMRP-CLIP, the FMRP-CLIP scores for the dendritic FMRP targets were significantly larger than the scores for the full set of dendrite-enriched mRNAs (*Figure 6D*). These data suggest that dendritic FMRP targets are a subset of previously identified FMRP targets. Interestingly, we identified a number of experimentally defined dendritic FMRP targets that had low levels of whole-cell FMRP cell binding (i.e., had negative CA1 FMRP CLIP scores, *Figure 6D*), indicating that these mRNAs are significantly more FMRP-bound in dendrites than in cell bodies.

## Subcellular compartment-specific FMRP-CLIP scores reveal functionally distinct groups of FMRP targets

Many directly bound FMRP target transcripts encode proteins that are implicated in autism spectrum disorders (ASDs) (*Darnell et al., 2011*; *Iossifov et al., 2012*; *Zhou et al., 2019*). We hypothesized that FMRP may regulate functional subsets of its targets in a subcellular compartment-specific manner, a phenomenon that would be reflected by differences in compartment-specific FMRP binding. To test this, we segregated all whole-cell CA1 FMRP CLIP targets according to their function by module detection using the HumanBase software (*Krishnan et al., 2016*). Eight functional modules were detected, three of which contained more than 100 genes (*Figure 7A*, *Supplementary file 1I*). The FM1 cluster, which contains 393 genes, is highly enriched for genes involved in nuclear regulation of gene expression, with the top GO terms being chromatin organization and modification and histone modification. FM2 (292 genes) is enriched for genes involved in ion transport and receptor signaling. The FM3 cluster (203 genes) contains genes involved in the maintenance of cell polarity and autophagy (*Figure 7B*).

To determine if any of these functional modules might be differentially regulated by FMRP in the dendrites and cell bodies of CA1 neurons, we performed gene set enrichment analysis (GSEA). We estimated enrichment of the FM1-3 transcripts among all FMRP-bound, CA1-expressed transcripts ranked by their dendritic or cell bodies-specific FMRP CLIP score. FM2 and FM3 clusters were highly enriched in FMRP-bound mRNAs in both the dendrites and the cell bodies, while the FM1 cluster was strongly enriched among cell body-bound FMRP targets, but only weakly enriched among the dendritic FMRP-bound transcripts (*Figure 7C*). This suggests that FM2 and FM3 modules contain mRNAs that are directly bound and regulated by FMRP in dendrites, and the FM1 cluster contains highly bound FMRP targets in the cell bodies, indicating distinct, biologically coherent regulation.

We further utilized compartment-specific FMRP-CLIP scores to identify functional modules of ASD candidate mRNAs subject to compartment-specific FMRP regulation (*Figure 7D*, *Figure 7—figure supplement 1A and B*, *Supplementary file 1J*). One module, AM2, contains transcripts enriched for glutamate signaling, learning, and memory, and is bound by FMRP in both the dendrites and cell

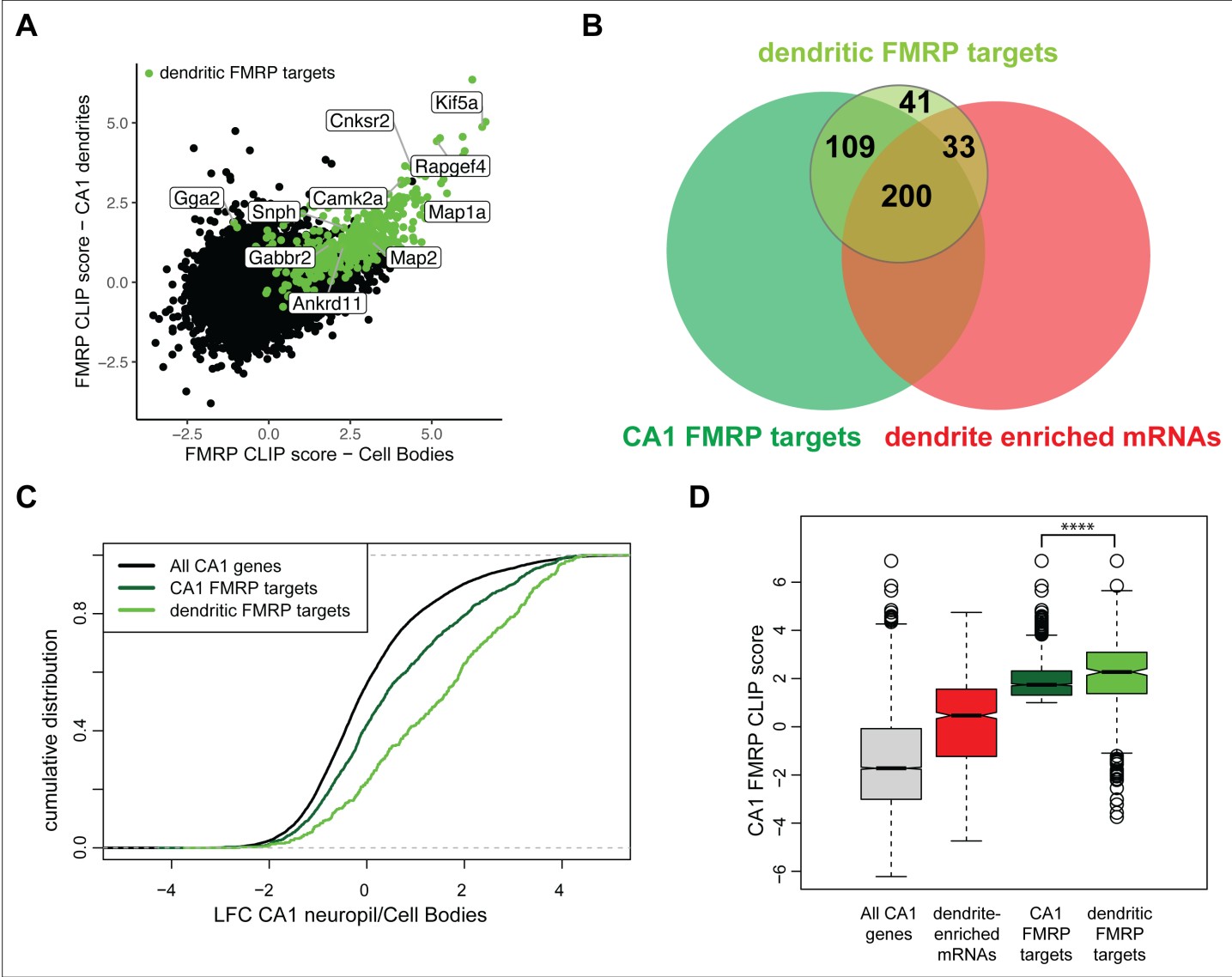

**Figure 6.** Compartment-specific cTag FMRP-CLIP reveals dendritic FMRP targets. (**A**) Compartment-specific Camk2a-cTag FMRP-CLIP and TRAP-seq were integrated to determine compartment-specific FMRP CLIP scores. CLIP scores were determined for all replicates. Plotted is the mean CLIP scores for the CA1 cell bodies and dendrites. Dendritic FMRP targets are colored in green. Genes of interest are labeled. (**B**) A subset of CA1 FMRP-CLIP targets (previously defined [*Sawicka et al., 2019*], dark green) were identified as dendritic FMRP-CLIP targets (light green). These are overlapped with dendrite-enriched mRNAs (*Figure 1B*) and whole-cell CA1 FMRP targets. (**C**) Dendritic FMRP targets are abundant in dendrites when compared to cell bodies. Dendrite-enrichment (log2 fold change [LFC]/standard error [SE] CA1 neuropil TRAP/cell bodies-TRAP) is plotted for all CA1 genes, all CA1 FMRP targets, and dendritic FMRP-CLIP targets. (**D**) Dendritic FMRP targets have high whole-cell FMRP-binding scores. Whole-cell CA1 FMRP CLIP scores (*Sawicka et al., 2019*) are plotted for all CA1 mRNAs, dendrite-enriched mRNAs, all CA1 FMRP targets and dendritic FMRP targets. Asterisks indicate significance in Wilcoxon rank-sum tests (****p<0.00001).

The online version of this article includes the following figure supplement(s) for figure 6:

**Figure supplement 1.** Compartment-specific cTag-FMRP-CLIP.

**Figure supplement 2.** Compartment-specific FMRP-CLIP scores.

bodies. The AM1 module consists of genes involved in chromatin modification and is highly enriched among mRNAs bound by FMRP in the cell bodies, but is not significantly enriched among dendritic FMRP-bound mRNAs. Taken together these observations suggest the possibility of compartmentalized roles for FMRP, in which mRNAs important for synaptic signaling are bound and regulated by FMRP near the synapses, while mRNAs bound by FMRP in the cell bodies are involved in the regulation of neuronal gene expression through chromatin regulation.

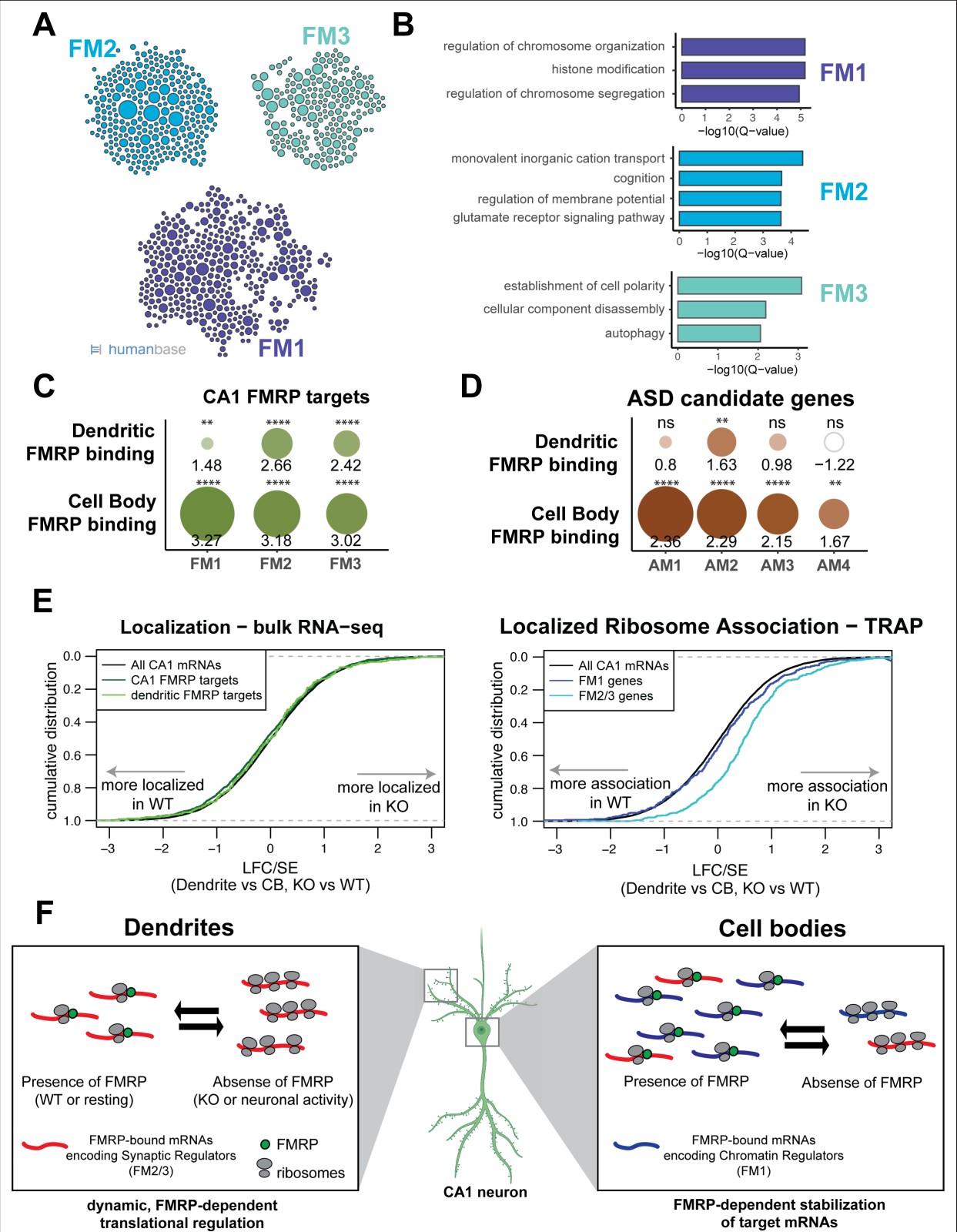

**Figure 7.** FMRP regulates functionally distinct mRNAs in the cell bodies and dendrites of CA1 neurons. (**A**) Whole-cell *CA1 FMRP* targets fall into three functional clusters. Functional module detection was performed for CA1 FMRP targets by the HumanBase software. (**B**) Top GO terms for the three largest functional modules of CA1 FMRP targets. Q-values for enrichment of terms were determined by the HumanBase software. (**C**) Dendritic FMRP targets are enriched in functionally distinct modules of CA1 FMRP targets. CA1 genes were ranked according to FMRP dendritic and cell bodies FMRP-

*Figure 7 continued on next page*

*Figure 7 continued*

CLIP scores, and gene set enrichment analysis (GSEA) was performed using the FMRP functional clusters (from **A**) as gene sets. Circles are colored according to normalized enrichment scores (NES) and sized according to false discovery rate (FDR) from the GSEA. NES values are shown, and asterisks indicate significance (\*\*FDR <0.001, \*\*\*\*FDR<0.00001). (**D**) Dendritic FMRP targets are enriched in a functional module of autism candidate genes. GSEA was performed as shown in (**C**), with functional modules of autism candidate genes (SFARI) clustered according to the HumanBase software. (**E**) Localization of FM2/3 FMRP targets is largely unchanged in compartment-specific bulk RNA-seq of FMRP KO animals, but increased in TRAP. Left: neuropil localization (log2 fold change [LFC]/standard error [SE] of CA1 neuropil bulk RNA-seq vs. cell bodies bulk RNA-seq) was assessed in FMRP KO vs. WT animals. Cumulative distribution plots are shown. Shifts to the right indicate more localization in the FMRP KO animals, and shifts to the left indicate more localization in WT animals. All CA1-expressed genes, all CA1 FMRP targets and dendritic FMRP targets are shown. Right: localized ribosome association in TRAP-seq on FMRP KO vs. WT animals, with subsets including the FM1/2/3 groups of CA1 FMRP targets as described in (**A**). (**F**) Distinct, compartment-specific FMRP regulation of functionally distinct subsets of mRNAs in CA1 cell bodies and dendrites. Localization of mRNA to the dendrites does not appear to be FMRP-dependent in CA1 neurons, but likely depends on other factors (e.g., other RNA-binding proteins or mRNA characteristics such as GC content, length, or secondary structure) that target mRNAs to the dendrite or compartments within the neuronal soma. In dendrites, the absence of FMRP increases the ribosome association of its targets; this finding is consistent with a model in which FMRP inhibits ribosomal elongation and thereby translation (*Darnell et al., 2011*). In resting neurons, the translation of FMRP-bound mRNAs encoding synaptic regulators (FM2 and FM3 mRNAs) is repressed. When FMRP is not functioning, due to either genetic alteration (FMRP KO or FXS) or neuronal activity-dependent regulation (e.g., FMRP calcium-dependent dephosphorylation; *Lee et al., 2011*; *Bear et al., 2004*), ribosome association and translation of targets are increased. In cell bodies, FMRP binds mRNAs that encode for chromatin regulators (the FM1 cluster of FMRP targets), as well as FM2/3 mRNAs (consistent with synapses forming on the cell soma). FM1 targets show patterns of mRNA regulation similar to what our group observed in bulk CA1 neurons: FMRP target abundance is decreased in FMRP KO cells, perhaps due to loss of FMRP-mediated block of degradation of mRNAs with stalled ribosomes (*Sawicka et al., 2019*; *Darnell, 2020*). This does not preclude the observation that FMRP also inhibits translation of chromatin regulators (*Korb et al., 2017*).

The online version of this article includes the following figure supplement(s) for figure 7:

**Figure supplement 1.** Predicted compartment-specific FMRP regulation of autism spectrum disorder (ASD) candidate genes.

**Figure supplement 2.** Dendritic localization of FMRP targets is unaffected in FMRP KO animals.

**Figure supplement 3.** Differential localization of 3′UTR isoforms is unaffected in FMRP KO animals.

**Figure supplement 4.** FMRP target RNAs are downregulated in the cell bodies and upregulated in the dendrites of CA1 neurons.

## FMRP regulates the ribosome association of its targets in dendrites

To better understand FMRP-dependent regulation of dendritic mRNAs, we examined the dendritic ribosome-bound transcriptome in FMRP KO animals. We performed bulk RNA-seq and cell-type-specific TRAP on microdissected hippocampi from WT and FMRP KO littermates. Bulk RNA-seq of microdissected material in FMRP WT and KO mice showed no overall change in the localization of FMRP targets (*Figure 7E*, left panel). In addition, the mRNAs found to be dendrite-present and dendrite-enriched in KO animals (as in *Figure 1*, *Supplementary file 1K*) show large overlap with those in WT animals (*Figure 7—figure supplement 2A and B*). We validated this finding by FISH in FMRP KO mouse brain slices and found no evidence for altered localization of FMRP targets into the neuropil (*Figure 7—figure supplement 2C–E*). Global analysis of 3′UTR usage differences in TRAP between dendrites and cell bodies in FMRP KO and WT animals also showed no significant (FDR < 0.05) instances of dysregulated localization of 3′UTR isoforms in FMRP KO animals (*Figure 7—figure supplement 3A*). We validated this finding using FISH in FMRP KO mice, which revealed no differences in isoform localization for *Cnksr2* or *Anks1b* mRNAs (*Figure 7—figure supplement 3B and C*).

Although the identities of the dendritic mRNAs found in FMRP WT and KO mice were similar, quantitative analysis of TRAP revealed that dendritic mRNA levels of ribosome-associated FMRP targets were increased in CA1 dendrites of KO mice (*Figure 7E*, right panel). Interestingly, this was evident for FM2/3, but not FM1 transcripts. While FMRP targets are generally downregulated in TRAP from hippocampal neurons (*Sawicka et al., 2019*), a finding that we replicate in cell bodies (*Figure 7—figure supplement 3A*), transcripts that encode synaptic regulatory proteins (FM2/3), which are bound by FMRP in the dendrites, show increased ribosome association in CA1 dendrites of KO animals (*Figure 7—figure supplement 3B*). These results suggest a model in which FMRP differentially regulates translation of functionally distinct mRNAs in specific neuronal compartments (see model in *Figure 7F*).

## Discussion

Recent advances in cell-type-specific transcriptomic approaches have greatly increased the resolution at which we understand gene expression in the nervous system. Here, we build on these advances

by incorporating compartment-specific CLIP and TRAP in order to define (1) a high-quality, cell-type-specific transcriptome of CA1 neuronal cell bodies and dendrites in vivo, (2) FMRP-bound mRNAs in dendrites, and (3) compartment-specific FMRP regulation of its targets. We found subcellular differences in the sets of alternatively spliced or polyadenylated transcripts in each compartment, connecting pre-mRNA nuclear regulation to subcellular localization in neurons (*Figures 2 and 3*). Moreover, previously defined, directly bound FMRP targets are overrepresented in the dendritic transcriptome, and FMRP preferentially binds to these dendritic mRNA isoforms. We find that the ribosome association of dendritic FMRP targets is increased in FMRP-null mice, consistent with differential translational regulation between subcellular compartments. Distinct sets of FMRP-bound autism-related transcripts have been described, particularly those related to chromatin regulation and synaptic plasticity (*Darnell et al., 2011*; *Iossifov et al., 2012*). Remarkably, we find here that these transcripts show different subcellular localization: transcripts encoding chromatin regulators are enriched in CA1 cell bodies, while those encoding synaptic regulators are enriched in their dendrites. Together, these observations indicate that RNA regulatory factors link post-transcriptional controls with local translation of RNA isoforms in neurons. The data support and extend a model (*Darnell, 2020*) in which FMRP integrates cellular activity and signaling to maintain neuronal homeostatic plasticity (*Turrigiano, 2012*) by mediating differential translation of transcripts encoding nuclear and synaptic functions in the cell body and dendrite, respectively (*Figure 7F*).

## The CA1 dendritic transcriptome

Much effort has been put into molecular profiling of the localized transcriptome, translatome, and proteome using in vitro neuron or neuron-like cell models (*Zappulo et al., 2017*; *Goering et al., 2020*; *Middleton et al., 2019*; *Taliaferro et al., 2016*). In vivo systems, such as microdissection of hippocampal CA1 regions, offer the advantage of profiling neurons that have formed physiological levels of relevant connections with surrounding neurons. Although RNA-sequencing (*Cajigas et al., 2012*), 3-Seq (*Tushev et al., 2018*), and TRAP-seq (*Ainsley et al., 2014*) have been performed previously for microdissected CA1 neuropil, these studies were either not performed using cell-type-specific approaches or unable to capture full-length mRNAs in resting neurons. As the mRNAs presented here are intact and relatively free of contaminating cell types (*Figure 1—figure supplement 1*), this dataset can be used for definition of dendritic ribosome-bound mRNAs and for identification of differential usage of 3′UTRs (*Figure 2*) and alternative splice isoforms (*Figure 3*) in CA1 neuropil and cell bodies compartments, making it a valuable dataset for the community.

Consistent with prior reports (*Tushev et al., 2018*), we find that the majority of differentially localized 3′UTRs are longer than their sequestered counterparts, suggesting that APA events that lead to longer 3′UTR isoforms might allow inclusion of localization and regulatory elements, such as binding sites for RNA-binding proteins or AGO-miRNA complexes. Long 3′UTRs may also act to recruit binding partners for nascent proteins, which can affect the function and/or localization of the protein, as previously reported (*Berkovits and Mayr, 2015*). Future experiments analyzing compartment-specific cTag-CLIP of RNA-binding proteins that bind to 3′UTRs such as AGO, Staufen, NOVA1/2, or ELAVL2/3/4 will provide further insight into the role of these 3′UTRs in mRNA localization and local translation.

In addition to a role for 3′UTR-APA in RNA localization and regulation, we find that 20% of differentially localized 3′UTRs result from APA events that impact the CDS. This finding underscores the possibility that differential mRNA localization may be linked to expression of functionally distinct protein isoforms generated during nuclear processing. This is further supported by our observation of AS events that result in differentially localized mRNA alternatively spliced isoforms, which has not been reported previously. We found that NOVA2, a neuron-specific splicing factor, is responsible for the generation of splicing isoforms that are sequestered to the neuronal cell bodies of CA1 neurons (*Figure 3E*). NOVA1 and NOVA2 are examples of a relatively small number of mammalian splicing factors demonstrated to directly bind to pre-mRNA and thereby regulate AS (*Licatalosi et al., 2008*; *Zhang and Darnell, 2011*) and also bind 3′ UTRs of those same transcripts (*Eom et al., 2013*). For example, in the case of GlyRa2, NOVA proteins co-localize with the transcript in the nucleus to regulate exon 3A splicing and in neuronal dendrite (*Racca et al., 2010*). These findings further underscore the many ways in which RNA-binding proteins contribute to neuronal complexity in specific subcellular compartments.

## FMRP binds dendritic mRNAs

The significant overlap between CA1 FMRP targets and dendrite-enriched mRNAs supports literature indicating that FMRP regulates a significant portion of the dendritic transcriptome (*Bagni and Zukin, 2019*; *Banerjee et al., 2018*; *Liu-Yesucevitz et al., 2011*). We do not find a role for FMRP in the localization of its targets in CA1 neurons, as demonstrated in our comparison of FMRP WT and KO brain. However, overall FMRP-binding affinity (defined by FMRP-CLIP scores in hippocampal neurons) correlates with relative dendritic abundance of a given mRNA (i.e., the enrichment of mRNAs in the neuropil over the cell body, *Figure 4G*), indicating a strong preference for FMRP binding on dendritic mRNA isoforms. This suggests the possibility that FMRP and localized mRNAs may be co-transported into the dendrites. It is also possible that FMRP may play a role in localization of its targets in other neuronal cell types, as has been suggested in radial glia from developing mouse brains (*Pilaz et al., 2016*). Future studies dissecting compartment-specific regulation of FMRP targets in cell-type systems such as these will be very informative.

Interestingly, through analysis of whole-cell cTag FMRP-CLIP data, we find multiple instances of FMRP selectively binding to specific dendritic isoforms (*Figure 5*). A striking example is the case of the *Cnksr2* gene, which generates a short, sequestered mRNA and a longer, highly localized isoform. The protein encoded from the dendritic mRNA isoform contains an additional PDZ-binding domain that is not present in the shorter isoform. *Cnksr2* has been identified in genome-wide association studies as an ASD candidate, and mutations in this gene have been shown to cause epilepsy and intellectual disability (*Aypar et al., 2015*). In the cell body compartment, the shorter isoform is predominant, which can be seen by both PAPERCLIP and TRAP. However, FMRP-CLIP, which generally binds the CDS and at least the proximal 3'UTR regions of its targets, shows predominant binding on the 3'UTR of a minor isoform in the presence of a more highly expressed, shorter, sequestered mRNA isoform (*Figure 5B*). Similar trends can be seen with a number of other mRNAs such as *Ankrd11* (*Figure 5*) and *Anks1b* (data not shown). Taken together, these data indicate that FMRP can display binding preferences both on different transcripts and different isoforms generated from a single gene. This finding adds an additional layer to the already-complicated process of how FMRP recognizes and binds its targets and suggests that FMRP binding specificity may rely on localization-determining events in the nucleus, such as deposition of RNA-binding proteins on the 3'UTRs of alternatively spliced transcripts.

FMRP binding to dendritic mRNA isoforms may also be a result of events that occur in the cytoplasm. For example, some mRNAs with longer 3'UTRs themselves may possess great propensity for entrance into FMRP-containing transport granules due simply to length. This would be consistent with observations that long mRNAs are preferentially found in stress granules due to lower translation efficiency and increased ability for RNA-RNA interactions to form, which are thought to stabilize RNA granules (*Khong et al., 2017*). This suggestion is also supported by findings that FMRP is found in neuronal mRNA transport granules (*Dictenberg et al., 2008*) and is known to bind to RNA structural elements such as kissing complexes and G-quadruplexes (*Darnell et al., 2005*), and support suggestions for a role for FMRP in maintaining the translationally repressed status of long mRNAs in transport granules.

Although previous work has proposed a role for the interaction between G-quadruplexes and the RGG domain of the FMRP protein in mRNA localization in an in vitro system (*Goering et al., 2020*), we did not find enrichment of G-quadruplexes in the 3'UTRs of dendritic FMRP targets when compared to dendritic non-FMRP targets. This is consistent with previous findings (*Sawicka et al., 2019*; *Darnell et al., 2011*) that direct FMRP binding occurs primarily on CDS and proximal 3'UTR portions of its targets without observable sequence specificity. This discrepancy could be the result of cell-type-specific functions of FMRP or may indicate that FMRP-directed regulation of G-quadruplex-containing mRNAs is not the result of stable binding of FMRP to these sequences.

We present here a list of mRNAs that are highly bound to FMRP in the dendrites of CA1 neurons. These 383 targets are significantly enriched in the dendrites and were found to have high FMRP CLIP scores in whole CA1 neurons (*Sawicka et al., 2019*, *Figure 6*), indicating higher than expected FMRP binding relative to other mRNAs of similar transcript abundance. Dendritic FMRP targets were determined by combining compartment-specific CLIP and TRAP experiments to determine compartment-specific FMRP CLIP scores. Importantly, these targets are the result of stringent filtering to include only high-confidence, experimentally defined dendritic FMRP targets.

Moreover, we bioinformatically extended our findings using compartment-specific FMRP CLIP scores to identify functional clusters of previously identified FMRP targets that are differentially abundant in the dendrites in respect to CA1 cell bodies. We find a remarkable link between the function of the protein product of a given FMRP target mRNA and its subcellular localization. The FM1 cluster, which contains FMRP target transcripts encoding proteins with nuclear functions such as histone modification and chromosome organization, is enriched in CA1 cell bodies. Approximately 10% of mRNAs encoding chromatin modifiers in CA1 neurons are FMRP targets. In contrast, FM2 and FM3 FMRP target mRNAs, which encode for proteins with synaptic functions such as ion transport, receptor signaling, and cell polarity, are found in both cell bodies and dendrites. Approximately 15–20% of CA1 mRNA encoding synaptic genes are members of the FM2 and FM3 clusters of FMRP targets. Together, these results indicate highly specific FMRP targeting of these two biologically coherent subgroups of targets in two distinct neuronal compartments, suggesting differential translation of chromatin transcripts in the cell body and synaptic-related transcripts in dendrites, and potentially where axonal synapses make contacts in the cell soma.

## Compartment-specific regulation of FMRP targets

Interestingly, mRNAs from genes in the FM2 and FM3 clusters show increased ribosome association in the FMRP KO mouse in a pattern distinct from the FM1 genes (*Figure 7E*). Bulk RNA-seq on the same compartments, as well as FISH on FMRP KO mouse brain, showed that overall FMRP targets levels were largely unchanged in abundance or localization in the neuropil. This suggests that in the absence of FMRP, while transcripts that are normally FMRP targets can still be localized to dendrites, they have an increased ribosome association. This supports the proposal (*Wang et al., 2008*) that FMRP in the neuronal processes may exist in a polyribosome-depleted granule, which is altered to become translationally competent upon neuronal activity. It is also consistent with the role of FMRP as a translational suppressor and detection of increased basal translation rates in mouse models of Fragile X syndrome (*Gross et al., 2010*; *Liu et al., 2012*).

Taken together, we suggest a model in which FMRP specifically binds mRNAs that encode synaptic proteins and are fated for dendritic localization and maintains them in a translationally repressed, and potentially polyribosome-depleted state for transport into the processes. Further, within the dendrite, our findings in FMRP-null mice are consistent with a role for neuronal activity to induce polyribosome formation and local translation (and concomitant increased polyribosome density) of its specific targets in dendrites. This may be through activity-dependent removal of FMRP from its targets, for example, by dephosphorylation (*Narayanan et al., 2007*, *Figure 7F*). Future experiments investigating how dendritic FMRP binding changes upon neuronal activity will help to elucidate the precise role of FMRP in regulation of activity-dependent local translation in dendrites.

We have previously shown in bulk CA1 neurons that FMRP target mRNAs are destabilized in the absence of FMRP (*Sawicka et al., 2019*). Earlier work also suggested that FMRP targets as a group are downregulated in the absence of FMRP (*Thomson et al., 2017*; *Ceolin et al., 2017*), and some evidence can be seen for translational activation of individual mRNAs. Our TRAP in CA1 cell bodies (*Figure 7—figure supplement 4A*) is consistent with this data, and with the proposal that the absence of FMRP leads to an overall decrease in steady-state mRNA abundance of its targets. We detect downregulation of both transcripts encoding chromatin regulators (FM1) and synaptic regulators (FM2/3) in the cell bodies in the absence of FMRP. We hypothesize that FMRP may act to protect mRNAs with stalled ribosomes from degradation, suggesting a role for FMRP in stabilization of translationally stalled mRNAs. A similar model has been proposed following the finding that the abundance of codon-optimized FMRP targets is decreased in FMRP KO (*Shu et al., 2020*; *Richter and Zhao, 2021*). It is reasonable to suspect that loss of FMRP may lead to an increase in translation in the cell body, as seen in other systems (*Greenblatt and Spradling, 2018*); however, TRAP does not allow for quantitation of ribosome occupancy, so we could not detect these changes using this method. Downregulation of steady-state mRNA levels in cell bodies in the absence of FMRP could also relate to homeostatic feedback on transcription (*Darnell, 2020*). However, in the dendrites we suggest that this pathway is either not present or is decreased in steady-state neurons, such that the absence of FMRP is seen as increased ribosome association of FMRP targets encoding synaptic regulators (FM2/3) and thus translational regulation of these mRNAs predominates in dendrites (*Figure 7F*).

An emerging theme in the study of FMRP is that not all targets are regulated in the same manner. Ribosome profiling and RNA-seq in FMRP KO cells in vitro identified distinct groups of FMRP targets whose localization and translation is regulated by the RGG- and KH- domains of the FMRP protein, respectively (*Goering et al., 2020*). Extensive ribosome profiling and RNA-seq in mouse brains showed functionally distinct groups of FMRP targets for which loss of FMRP leads to changes in either mRNA levels or translational efficiency (*Shah et al., 2020*). Our work suggests that subcellular localization of FMRP targets may be a critical factor in these distinct modes of FMRP-mediated regulation. Further, we present three functionally distinct clusters of CA1 FMRP targets and suggest that the cluster that contains chromatin regulators (FM1) are specifically regulated in the cell bodies, whereas synaptic regulators (FM2/3) are regulated in both compartments.

In summary, we demonstrate the ability to utilize compartment- and cell-type-specific RNA profiling technologies to precisely define the dendritic transcriptome. Our results underscore the role of FMRP as an important regulator of dendritic mRNAs, playing an important function in ribosome association of isoform-specific dendritic targets and local translational control. This finding, coupled with the identification of FM1 chromatin-associated transcripts regulated by FMRP exclusively in the cell bodies, supports the hypothesis (*Darnell, 2020*) that FMRP acts as a sensor for neuronal activity through actions on both neuronal transcription and synaptic activity. Further studies into how these subsets of mRNAs are differentially FMRP-regulated in a subcellular compartment-specific manner will have important implications in the understanding of how dysregulation of FMRP and its targets leads to intellectual disability and ASD.

# Materials and methods

**Key resources table**

| Reagent type (species) or resource | Designation | Source or reference | Identifiers | Additional information |
|---|---|---|---|---|
| Strain, strain background (*Mus musculus* C57BL6/J) | B6.Cg-Tg(Camk2a-cre)T29-1Stl/J | Jackson Laboratory | RRID:IMSR_JAX:005359 | Referred to as Camk2a-Cre |
| Strain, strain background (*M. musculus* C57BL6/J) | B6N.129-Rpl22tm1.1 Psam/J | Jackson Laboratory | RRID:IMSR_JAX:011029 | Referred to as RiboTag |
| Strain, strain background (*M. musculus* C57BL6/J) | B6.129P2-Fmr1tm1Cgr/J | Gift from W.T. Greenough | RRID:IMSR_JAX:003025 | Referred to as Fmr1 KO |
| Strain, strain background (*M. musculus* C57BL6/J) | Fmr1-cTag | *Sawicka et al., 2019* | | |
| Strain, strain background (*M. musculus* C57BL6/J) | cTag-PABP | PMID:28910620 | | |
| Antibody | NeuN, guinea pig polyclonal | Millipore | Millipore Cat# ABN90P; RRID:AB_2341095 | For IF (1:2000) |
| Antibody | Anti-HA tag, rabbit polyclonal | Abcam | Abcam Cat# ab9110; RRID:AB_307019 | For IP (20–80 µg/mL depending on region) |
| Antibody | Anti-HA tag, rabbit monoclonal | Cell Signaling | Cell Signaling Cat# C29F4; RRID:AB_1549585 | For IF (1:4000) |
| Antibody | Anti-GFP antibodies HtzGFP19C8 and HtzGFP19F7, mouse monoclonal | PMID:19013281 | Heintz Lab; Rockefeller University Cat# Htz-GFP-19C8; RRID:AB_2716737 Heintz Lab; Rockefeller University Cat# Htz-GFP-19F7; RRID:AB_2716736 | For IP (25 µg each antibody for 1.2 mL lysate prepared from 8 to 10 animals) |
| Antibody | Anti-BrdU, mouse monoclonal [IIB5] | Abcam | Abcam Cat# ab8955; RRID:AB_306886 | For IP (5 µg per pooled RT reaction) |

*Continued on next page*

*Continued*

| Reagent type (species) or resource | Designation | Source or reference | Identifiers | Additional information |
|---|---|---|---|---|
| Software, algorithm | Zen Black | Zeiss | RRID:SCR_018163 | |
| Software, algorithm | Imaris | Oxford Instruments | RRID:SCR_007370 | |
| Software, algorithm | featureCounts | PMID:24227677 | RRID:SCR_012919 | |
| Software, algorithm | DESeq2, Bioconductor | PMID:25516281 | RRID:SCR_015687 | |
| Software, algorithm | CLIP Tool Kit (CTK) | PMID:27797762 | | |
| Software, algorithm | goseq, Bioconductor | PMID:20132535 | RRID:SCR_017052 | |
| Software, algorithm | RSeQC | PMID:22743226 | RRID:SCR_005275 | |
| Software, algorithm | GenomicRanges, Bioconductor | PMID:23950696 | RRID:SCR_000025 | |
| Software, algorithm | Limma, Bioconductor | PMID:25605792 | RRID:SCR_010943 | |
| Software, algorithm | fgsea, Bioconductor | *Korotkevich et al., 2021* | RRID:SCR_020938 | |
| Software, algorithm | DEXSeq | PMID:22722343 | RRID:SCR_012823 | |
| Software, algorithm | rMATS | PMID:25480548 | | |
| Software, algorithm | HumanBase | Simons Foundation | RRID:SCR_016145 | |
| Commercial assay or kit | Quant-iT RiboGreen RNA Assay Kit | Thermo Fisher Scientific | Cat# R11490 | |
| Commercial assay or kit | High Pure RNA isolation kit | Roche | Cat# 11828665001 | |
| Commercial assay or kit | Dynabeads mRNA Purification Kit | Thermo Fisher Scientific | Cat# 61006 | |
| Commercial assay or kit | TruSeq RNA library prep kit | Illumina | Cat# RS-122-2001 | |
| Commercial assay or kit | RNAscope Multiplex Fluorescent Reagent Kit v2 | Advanced Cell Diagnostics | Cat# 323100 | |

## Mice

All mouse procedures were conducted according to the Institutional Animal Care and Use Committee (IACUC) guidelines at the Rockefeller University. RiboTag (B6N.129-Rpl22<sup>tm1.1Psam</sup>/J, stock no. 011029) and Camk2a-Cre (B6.Cg-Tg(Camk2a-cre)T29-1Stl/J, stock no. 005359) were obtained from Jackson Laboratories. FMRP cTag (*Sawicka et al., 2019*) and PABPC1 cTag (*Hwang et al., 2017*) mice were previously described. B6.129P2-Fmr1tm1Cgr/J (*Fmr1* KO) mice were a generous gift from W.T. Greenough maintained for multiple generations in our own facilities. Mice were housed up to five mice per cage in a 12 hr light/dark cycle. Breeding schemes for TRAP-seq (producing RiboTag$^{+/-}$, Fmr1$^{+/+}$, and RiboTag$^{+/-}$,Fmr1$^{Y/-}$ male littermates) and FMRP cTag-CLIP (producing Cre$^{+/-}$; Fmr1-cTag$^{+/Y}$ male offspring) were described previously (*Sawicka et al., 2019*).

## Immunofluorescence

Immunofluorescence was performed as described previously (*Sawicka et al., 2019*). Primary antibodies used were NeuN (Millipore ABN90P, RRID:AB_2341095, 1:2000 dilution) and HA (Cell Signaling, C29F4, RRID:AB_1549585, 1:4000 dilution).

## TRAP- and RNA-seq of microdissected hippocampal slices

For each TRAP-seq replicate (four replicates were performed), hippocampi from three adult mice (6–10 weeks) were sectioned into 300 µm slices using a tissue chopper and microdissected in HBSS containing 0.1 mg/mL cycloheximide. For microdissection, the CA1 was excised from the hippocampal slices and separated into a cell body (CB) and neuropil layer. Microdissected tissue from each mouse was collected and resuspended in 0.5 mL ice-cold polysome buffer (20 mM HEPES, pH 7.4, 150 mM NaCl, 5 mM MgCl$_2$, 0.5 mM DTT, 0.1 mg/mL cycloheximide) supplemented with 40 U/ml RNasin Plus

(Promega) and cOmplete Mini EDTA-free Protease Inhibitor (Roche) and homogenized by mechanical homogenization with 10 strokes at 900 rpm. NP-40 was added to 1% final concentration and incubated on ice for 10 min. Samples were pooled and centrifuged at 2000 × $g$ for 10 min. Supernatant was subsequently centrifuged at 20,000 × $g$ for 10 min. 10% of the resulting lysate was used for RNA-seq, and the remaining lysate was subject to pre-clearing with 1.5 mg (50 µL) Protein G Dynabeads for 45 min. HA-tagged ribosomes were collected by indirect IP by adding 40 µg of anti-HA antibody (Abcam ab9110, RRID:AB_307019) to CB lysate pools and 5 µg to NP lysate pools. IP was performed overnight with rotation at 4°C. Antibody-ribosome complexes were collected by addition of 7.2 mg (CB pools) or 4.44 mg (NP pools) Protein G Dynabeads and further incubated with rotation at 4°C for 1 hr. Beads were washed with 1 mL polysome buffer containing 1% NP-40 once for 5 min and twice for 20 min, followed by 4 × 10 min washes in 50 mM Tris pH 7.5, 500 mM KCl, 12 mM MgCl₂, 1% NP-40, 1 mM DTT, 0.1 mg/mL cycloheximide. RNA was extracted from beads by incubating in 500 µL Trizol at room temperature for 5 min. RNA was collected by standard Trizol (Invitrogen) extraction via the manufacturer's protocol and quantified with RiboGreen Quant-IT assays (Invitrogen). Bulk RNA-seq samples were treated with RQ1 RNase-free DNase (Promega) prior to library preparation. RNA was further purified for polyadenylated RNA by using Dynabeads mRNA Purification Kit (Ambion). The libraries were prepared by TruSeq RNA Sample Preparation Kit v2 (Illumina) following the manufacturer's instructions. High-throughput sequencing was performed on HiSeq (Illumina) to obtain 100 nucleotide paired-end reads.

## FISH with RNAscope

Mice were anesthetized with isoflurane and transcardially perfused with PBS containing 10 U/mL heparin followed by perfusion with ice-cold PBS containing 4% paraformaldehyde. After perfusion, animals were decapitated and intact brains removed and postfixed overnight in 4% paraformaldehyde in PBS at 4°C. Brains were then transferred to PBS with 15% sucrose for 24 hr followed by PBS with 30% sucrose for a further 24 hr and then embedded and frozen in OCT medium. 12 µm coronal slices were prepared using a Leica CM3050 S cryostat and directly adhered to Fisherbrand 1.0 mm superfrost slides (Cat # 12-550-15) and stored at –80°C until use. FISH was performed using the RNAscope Multiplex Fluorescent Kit v2 as recommended for fixed frozen tissue, with some exceptions. For pretreatment of samples prior to hybridization, slides were baked at 60°C for 45 min, followed by fixation in 4% paraformaldehyde in PBS at 4°C for 90 min. Samples were dehydrated in ethanol (50, 70, and 100% twice each) and incubated at room temperature before hydrogen peroxide treatment for 10–20 min, followed by target retrieval as recommended. After probe hybridization, samples were washed three times for 15 min in wash buffer heated to 37°C. Probes used were conjugated with Alexa fluorescein (488 nm), Alexa Cyanine 3 (555 nm), and Alexa Cyanine5 (647 nm). RNAscope probes were designed to recognize unique 3'UTR sequences (for UR-APA events) or for common and distal 3'UTRs (for UTR-APA events) with at least 500–1000 nts between regions. See *Supplementary file 1B*. Each FISH experiment was performed on at least three slices from at least two different mice.

## Image processing and quantitation

Airyscan-Fast (AS-F) image capturing was performed using the Zen Black 2.3 SP1 FP3 acquisition software on an Inverted LSM 880 Airyscan NLO laser scanning confocal Microscope (Zeiss) outfitted with AS-F module (16 detectors) and argon laser for 488 line. Objective: Zeiss Plan 63 × 1.4 NA Apochromat oil immersion; imaging at this objective was performed using Immersol 518F immersion media (ne = 1.518 [23°C]; Carl Zeiss). Acquisition parameters include laser lines 405 nm, 488 nm, 561 nm, and 633 nm (laser power adjusted until relative power for each line eliminates as much background as possible without diminishing signal). Emission filter for Airyscan detection: 405ch, BP 420–480+ BP 495–620; 488ch, BP 420–480 + 495-550; 561ch, BP 420–480 + 495-620; 633ch, BP 570–620+ LP645. Settings: eight bit-depth and acquired with image size: 135.0 × 135.0 µm; pixel size: 0.14 µm (step size is 0.159 using a piezo stage). All raw image data was sent directly to ZEN 2.3 software for reconstruction. Files underwent Airyscan processing (parameters: auto strength at 6 for 3D images) before being stitched at a normalized cross-correlation threshold set at 7. Processed and stitched .czi files were converted to .ims files using Imaris File Converter x64 9.6.0 before being uploaded into Imaris x64 9.6. Spots were quantified using the spot counting operation (Imaris software) with the default values and modifying the spot detection parameters ('Model PSF-elongation

along Z-axis': estimated XY diameter: 0.8 μm; estimated Z diameter: 1.4 μm). Detection threshold was adjusted manually until all false/weak signals were eliminated. The mRNA coordinates (X, Y, Z) were downloaded for bioinformatic analysis. Max projections exported from Imaris were uploaded in Fiji. Images were adjusted to eight-bit, orientation is adjusted, and channels are separated. For detection of nuclei for bioinformatic analysis, threshold was adjusted until the majority of the DAPI stain was detected and applied. 'Analyze particles' operation was applied with the settings size 50-infinity (pixel units); circularity 0.0–1.0; show 'masks.' Resulting text image files were used for downstream analysis.

### Compartment-specific cTag FMRP-CLIP

Microdissection of hippocampal slices from 5 to 8 adult Camk2a-FMRP-cTag mice was performed as described above, except that the slices were UV crosslinked in HBSS with 0.1 mg/mL cycloheximide three times using 400 mJ/cm$^2$ after sectioning and before microdissection. After dissection, samples were collected and homogenized in lysis buffer (1× PBS, 0.1% SDS, 0.5% NP-40, 0.5% sodium deoxycholate supplemented, 1X cOmplete Mini EDTA-free Protease Inhibitor [Roche] and 0.1 mg/mL cycloheximide) by passing through syringes with a 28 gauge needle. cTag FMRP-CLIP was performed as described previously (*Sawicka et al., 2019*), with minor modifications. Cell body pools were lysed in 1 mL of lysis buffer and neuropil pools in 0.5 mL. Pre-clearing was performed with 6 and 1.5 mg of Protein G Dynabeads for CB and NP pools, respectively. IP was performed using mouse monoclonal anti-GFP antibodies conjugated to Protein G Dynabeads using 25 μg of each antibody for CB pools and 6.25 μg of each antibody for NP pools and rotated at 4°C for 1–2 hr. IPs washes were rotated 2–3 min at room temperature. RNA tags were cloned as described previously (*Sawicka et al., 2019*), with cell bodies and neuropil samples being pooled after barcoding in order to increase yield for low-input samples.

### Compartment-specific cTag-PAPERCLIP

Collection and UV crosslinking of microdissected material was performed as described for compartment-specific cTag FMRP-CLIP. cTag-PAPERCLIP was performed as described previously (*Hwang et al., 2017*) with the following exceptions. Four replicates were performed, using 3–14 mice per replicate. CB pools were lysed in 1 mL of lysis buffer, NP pools in 0.5 mL. Additional IP washes were performed using stringent washes conditions (described in *Sawicka et al., 2019*), and low-input samples were pooled after barcoding. Cell body pools were lysed in 1 mL of lysis buffer and neuropil pools in 0.5 mL. IP was performed using mouse monoclonal anti-GFP antibodies conjugated to Protein G Dynabeads using 25 μg of each antibody for CB pools and 6.25 μg of each antibody for NP pools and rotated at 4°C for 3–4 hr. RNA tags were cloned as described previously (*Hwang et al., 2017*) with cell bodies and neuropil samples being pooled after barcoding in order to increase yield for low-input samples.

### Bioinformatics

#### Calling localized mRNAs

Transcript expression was quantified from RNA-seq and TRAP-seq using salmon and mm10 gene models. Pairwise comparisons with batch correction were performed using DESeq2 for CA1 neuropil vs. cell bodies, with and without Cre expression, and TRAP vs. bulk RNA-seq. Dendrite-localized genes were defined as those with a Benjamini–Hochberg FDR less than 0.05 for FDR for TRAP vs. RNA-seq, log2 fold change (LFC) TRAP vs. RNA-seq greater than 0, and LFC Cre-positive vs. Cre-negative greater than 0 (all in CA1 neuropil samples only). Dendrite-enriched mRNAs used the same filters, but also required an FDR of CA1 neuropil vs. cell bodies of less than 0.05. Dendritic localization is defined as the LFC resulting from DESeq2 analysis of CA1 neuropil vs. cell bodies TRAP samples. For length and GC content analysis, the transcript that showed the highest expression in whole-cell hippocampal Camk2a-TRAP (*Sawicka et al., 2019*) was used.

#### GO analysis

GO analysis was performed using the goseq R package (*Young et al., 2010*). Background lists used were all CA1-expressed mRNAs (*Figures 1C and 3C*) or all dendrite-enriched mRNAs (*Figure 4C*).

#### FISH quantification

Nuclei (from DAPI stains) and spots (from FISH) were identified and their locations in the image determined with Fiji and Imaris software. For prediction of the location of the cell body layer in each image,

nuclei and spot-containing pixels were identified and converted into scatterplots in R. Scatterplots were sliced into 25 vertical slices, and the density of each slice was plotted in order to identify the location of the bottom of the cell body layer in each slice. These points were subject to two rounds of polynomial curve fitting, with outliers removed manually between the two rounds. The predicted distance between each FISH spot and the cell body was determined using the distance between the spot and the fitted curve. For *t*-tests, spots were considered to be in the neuropil if they were more than 10 μmfrom the predicted line. Changes in distribution were also assessed using Kolmogorov–Smirnov tests. For *Figure 7—figure supplements 2 and 3*, 1000 spots were samples from each picture, and the spots were binned into 15 bins according to their distance from the cell bodies layer. For each bin, differences in the mean percent of spots found in these regions in WT vs. KO pictures were determined by *t*-tests.

## Identification of differentially localized 3′UTR isoforms

polyA sites were identified from PAPERCLIP data using the CTK package (*Shah et al., 2017*) as described previously. From whole-cell PAPERCLIP datasets, peaks were considered that had 10 or more tags and represented 5% or more of the tags on that gene. For microdissected PAPERCLIP datasets, any peaks that had tags in more than one neuropil PAPERCLIP experiment were considered. Splice junctions were identified in both whole-cell and micro-dissected TRAP samples. Splice junctions were considered if they were found in 10 reads or if they represented 10% of total junction reads for that gene. Using the GenomicRanges package (*Lawrence et al., 2013*), the upstream splice junction was identified for each PAPERCLIP site, and the downstream PAPERCLIP site was identified for each splice junction. Percentage of covered bases for these potential 3′UTRs was determined using bedtools (*Quinlan and Hall, 2010*) and only those with 80% coverage in any single experiment were considered in downstream analyses. Next, ambiguous genes and 3′UTRs that overlapped other genes/UTR were eliminated. This yields all expressed final exons. Genes with multiple 3′UTRs were selected and used for counting of reads from microdissected TRAP-seq samples using featureCounts, followed by DEXSeq analysis (*Anders et al., 2012*) to identify differentially localized 3′UTRs. For 3′UTR usage analysis in KO animals, the expressed 3′UTRs identified in *Figure 2* were used, and reads aligning to each 3′UTR in WT and KO cell bodies and CA1 neuropil TRAP were quantitated using featureCounts as above. Differences in 3′UTR usage between neuropil and cell bodies in KO vs. WT animals were determined using DEXSeq including both genotype and region into the model.

### Splicing

Splicing analysis was performed using rMATS (*Shen et al., 2014*), considering both junction counts and exon coverage and the maser R package was used for visualization. For splicing analysis of RNABP KO mice, rMATS analysis was performed on datasets shown in *Supplementary file 1F*. Sashimi plots were generated in IGV.

### Searching for G-quadruplexes

In order to identify CA1 mRNAs with experimentally determined G-quadruplexes, sequences from K+-dependent G-quadruplexes from *Guo and Bartel, 2016* were found within the 3′UTR sequences of CA1 mRNAs (defined from TRAP; *Sawicka et al., 2019*). To find G-quadruplex motifs, the regular expression "[AU]GGA(.{0,6})[AU]GGA(.{0,6})[AU]GGA(.{0,6})[AU]GGA" was searched for in the 3′UTR of CA1 mRNA sequences using the gregexpr function in R.

### Compartment-specific CLIP

CLIP tags were processed as described previously (*Sawicka et al., 2019* for FMRP-CLIP and *Hwang et al., 2017* for cTag-PAPERCLIP). Briefly, for FMRP-CLIP, tags were mapped to the transcriptome using the transcript with the highest expression for each gene as determined by whole-cell Camk2a-TRAP (*Sawicka et al., 2019*). For cTag-PAPERCLIP, tags were mapped to the genome and polyadenylation sites were determined by clusters called using the CTK software (*Shah et al., 2017*).

## Calling dendritic FMRP targets

Counts of FMRP-CLIP tags mapped to transcripts were normalized first for transcript length and then by sequencing depth (scaled to 10,000 tags) in order to generate length and library size normalized CLIP expression values for each transcript. mRNAs were determined to be dendritic FMRP targets if they fit one of two criteria: (1) if they were reproducibly detected in cTag-FMRP-CLIP on the neuropil (>5 normalized tags per 10,000 in at least 3 of 5 replicates, 287 genes) or (2) if they had a mean compartment-specific CLIP score >1 (241 genes). See *Supplementary file 1G* for CLIP scores and CLIP expression information. CLIP scores were determined as described previously (*Sawicka et al., 2019*), with a few exceptions to account for low numbers of dendritic CLIP tags. All CLIP tags that map along the length of CA1 mRNAs were used for analysis. CLIP expression scores were calculated by dividing CLIP tags by transcript length, followed by normalization for library depth. TPMs for TRAP-seq were determined by the tximport package from pseudocounts obtained from salmon (*Patro et al., 2017*; *Soneson et al., 2015*). For each CLIP replicate and compartment, TRAP TPMs were plotted against CLIP expression scores with a TRAP TPM > 1 and FMRP-CLIP tags in three or more replicates. Linear models were determined, and mean CLIP scores were calculated as described previously (*Sawicka et al., 2019*).

## Functional clustering of FMRP targets

Functional module detection implemented within the HumanBase software was used to determine functional clusters of previously defined CA1 FMRP targets (https://hb.flatironinstitute.org/module/). Compartment-specific FMRP CLIP scores were determined essentially as described above, except without filtering for reproducibly detected mRNAs in order to maximize the number of genes included in the analysis. For GSEA, CA1 mRNAs were ranked by compartment-specific FMRP CLIP scores. GSEA was performed using the fgsea package (*Korotkevich et al., 2021*) using the gene lists from module detection as pathways.

# Acknowledgements

The authors wish to thank Alison North and Tao Tong from the Rockefeller University Bio-Imaging Resource Center for help with microscopy and image analysis, the Bioinformatic Resource Center at Rockefeller University for bioinformatics advice and support, and members of the Darnell lab for manuscript review. This work was supported by an award from the Leon Levy Foundation for Mind, Brain and Behavior to CRH, the Simons Foundation Research Award DFARI 240432 and NIH Awards NS081706 and R35NS097404 to RBD. RBD is an Investigator of the Howard Hughes Medical Institute.

# Additional information

## Funding

| Funder | Grant reference number | Author |
| --- | --- | --- |
| Leon Levy Foundation | | Caryn R Hale |
| Simons Foundation | | Caryn R Hale<br>Kirsty Sawicka<br>Kevin Mora<br>John J Fak<br>Jin Joo Kang<br>Paula Cutrim<br>Robert B Darnell |
| National Institute of General Medical Sciences | R35NS097404 | Caryn R Hale<br>Kirsty Sawicka<br>Kevin Mora<br>John J Fak<br>Jin Joo Kang<br>Paula Cutrim<br>Robert B Darnell |

| Funder | Grant reference number | Author |
|---|---|---|
| Howard Hughes Medical Institute | | Robert B Darnell |
| National Institutes of Health | NS081706 | Caryn R Hale<br>Kirsty Sawicka<br>Kevin Mora<br>John J Fak<br>Jin Joo Kang<br>Paula Cutrim<br>Robert B Darnell |

The funders had no role in study design, data collection and interpretation, or the decision to submit the work for publication.

## Author contributions

Caryn R Hale, Conceptualization, Formal analysis, Investigation, Writing – original draft; Kirsty Sawicka, Conceptualization, Investigation, Writing - review and editing; Kevin Mora, Data curation, Formal analysis, Investigation, Methodology, Writing - review and editing; John J Fak, Paula Cutrim, Katarzyna Cialowicz, Data curation, Methodology; Jin Joo Kang, Investigation, Methodology; Thomas S Carroll, Formal analysis, Methodology, Writing - review and editing; Robert B Darnell, Conceptualization, Funding acquisition, Investigation, Writing - review and editing

## Author ORCIDs

Kirsty Sawicka (iD) http://orcid.org/0000-0003-4195-6327
Robert B Darnell (iD) http://orcid.org/0000-0002-5134-8088

## Ethics

All mouse procedures were conducted according to the Institutional Animal Care and Use Committee (IACUC) guidelines at the Rockefeller University using protocol numbers 14678, 17013 and 20028.

## Decision letter and Author response

Decision letter https://doi.org/10.7554/eLife.71892.sa1
Author response https://doi.org/10.7554/eLife.71892.sa2

# Additional files

## Supplementary files

• Supplementary file 1. Tables used in this study. (A) Dendrite-present mRNAs results of DESeq2 analysis. 'Neuropil_vs_CB' indicates results of CA1 neuropil/CA1 cell bodies TRAP comparison, 'TRAP_vs_RNAseq' indicates results of CA1 neuropil TRAP/CA1 neuropil RNAseq comparison. 'CrePos_vs_CreNeg' indicates results of Camk2a-Cre-positive CA1 neuropil TRAP/Camk2a-Cre-negative CA1 neuropil TRAP comparison. (B) Dendrite-enriched mRNAs results of DESeq2 analysis. 'Neuropil_vs_CB' indicates results of CA1 neuropil/CA1 cell bodies TRAP comparison, 'TRAP_vs_RNAseq' indicates results of CA1 neuropil TRAP/CA1 neuropil RNAseq comparison. 'CrePos_vs_CreNeg' indicates results of Camk2a-Cre-positive CA1 neuropil TRAP/Camk2a-Cre-negative CA1 neuropil TRAP comparison. (C) RNAscope probes used in this study. (D) DEXSeq analysis of differentially localized 3′UTR isoforms. Results of DEXSeq analysis comparing 3′UTR expression in CA1 neuropil and cell bodies TRAP is shown. APA event types are indicated (APA type), and the relative location (UTR type) of the 3′UTR-APA UTRs is indicated. Genomic coordinates for mm10 are given. (E) rMATs analysis of differentially localized AS events. Results of rMATs analysis comparing alternative splice events from CA1 neuropil vs. cell bodies TRAP. Results are organized by event type, and coordinates of the exons involved in the splice event are given. (F) RNABP KO data used for splicing analysis. Description of RNA-seq data used for analysis of RNABP KO data in mouse brain. (G) Compartment-specific FMRP-CLIP scores. All CA1-expressing genes with detectable FMRP binding in the CA1 cell bodies and dendrites were assigned a compartment-specific FMRP-CLIP score. (H) dendritic FMRP targets. (I) FMRP target functional clusters. Whole-cell FMRP targets were divided into functional clusters by the HumanBase software. Three clusters had >100 genes. The gene names for those clusters are shown here. (J) ASD SFARI genes functional clusters. Potential ASD genes (from SFARI) were divided into functional clusters by the HumanBase software. Four clusters had >100 genes. The gene names for those clusters are shown here. (K) DESeq2 analysis for

FMRP KO vs. WT TRAP. Results of DESeq2 analysis. 'Localization' indicates results of CA1 neuropil/ CA1 cell bodies TRAP, FMRP KO/WT comparison, 'CB_only' indicates results of CA1 cell bodies TRAP only, FMRP KO/WT. 'Neuropil_only' indicates results of CA1 neuropil TRAP only, FMRP KO/ WT comparison.

• Transparent reporting form

## Data availability

Sequencing data have been deposited in GEO under accession code GSE174303, https://www.ncbi. nlm.nih.gov/geo/query/acc.cgi?acc=GSE174303.

The following dataset was generated:

| Author(s) | Year | Dataset title | Dataset URL | Database and Identifier |
|---|---|---|---|---|
| Hale CR, Darnell RB, Carroll T | 2022 | Cell-type and compartment-specific TRAP-seq, RNAseq, FMRP-CLIP, and PAPERCLIP from mouse microdissected CA1 | https://www.ncbi. nlm.nih.gov/geo/ query/acc.cgi?acc= GSE174303 | NCBI Gene Expression Omnibus, GSE174303 |

The following previously published datasets were used:

| Author(s) | Year | Dataset title | Dataset URL | Database and Identifier |
|---|---|---|---|---|
| Vuong C, Lin C, Black LD | 2018 | RbFox1-Nestin hippocampus | https://www.ncbi. nlm.nih.gov/geo/ query/acc.cgi?acc= GSE96722 | NCBI Gene Expression Omnibus, GSE96722 |
| Lovci MT | 2013 | Rbfox proteins regulate alternative mRNA splicing through evolutionarily conserved RNA bridges | https://www.ncbi.nlm. nih.gov/bioproject/ PRJNA215942 | NCBI BioProject, PRJNA215942 |
| Huang H | 2016 | RBFOX3/NeuN is required for hippocampal circuit balance and function | https://www.ncbi. nlm.nih.gov/geo/ query/acc.cgi?acc= GSE84786 | NCBI Gene Expression Omnibus, GSE84786 |
| Columbia University Medical Center | 2018 | Accession: PRJNA453385 ID: 453385 Precise temporal regulation of alternative splicing during neural development | https://www.ncbi.nlm. nih.gov/bioproject/ PRJNA453385/ | NCBI BioProject, PRJNA453385 |
| Saito Y | 2016 | Nova HITS-CLIP and RNA-Seq in mouse cortex | https://www.ncbi. nlm.nih.gov/geo/ query/acc.cgi?acc= GSE69711 | NCBI Gene Expression Omnibus, GSE69711 |
| Hwang H, Darnell RB | 2017 | cTag-PAPERCLIP Reveals Alternative Polyadenylation Promotes Cell-Type Specific Protein Diversity and Shifts Araf Isoforms with Microglia Activation | https://www.ncbi. nlm.nih.gov/geo/ query/acc.cgi?acc= GSE94054 | NCBI Gene Expression Omnibus, GSE94054 |
| Sawicka K, Darnell RB, Darnell JC | 2020 | The Fragile X protein, FMRP, has a cell-type specific role in CA1 hippocampal neurons to regulate transcripts encoding autism-spectrum proteins and circadian memory | https://www.ncbi. nlm.nih.gov/geo/ query/acc.cgi?acc= GSE127847 | NCBI Gene Expression Omnibus, GSE127847 |

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
