## [Editor Report]

The authors performed transcriptomic analyses from compartment-specific, micro-dissected hippocampal region tissue from transgenic mice. One feature that distinguishes this work from previous FMRP studies is the use of conditional knock-in tags (GFP or HA) and tissue-specific expression of the Cre recombinase to target a population of pyramidal neurons in the CA1 region. The strengths of the paper are the rich datasets and innovative integration of methods that will provide a valuable technical resource for the field.

---

## [Decision Letter]

**Decision letter after peer review:**

Thank you for submitting your article "FMRP regulates mRNAs encoding distinct functions in the cell body and dendrites of CA1 pyramidal neurons" for consideration by *eLife*. Your article has been reviewed by 2 peer reviewers, and the evaluation has been overseen by Robert Singer as Reviewing Editor and James Manley as the Senior Editor. The following individual involved in review of your submission has agreed to reveal their identity: Young Yoon (Reviewer #2).

Essential revisions:

Two key experiments are needed to support the claim that FMRP plays a role in localization, which is only correlative from the data shown. First is to perform quantitative analysis of FMRP target mRNA localization in dendrites from WT vs. Fmr1 KO mice. A second additional key validation experiment is to show that 1 or 2 mRNAs with high levels of FMRP-CLIP reads, for example, specific 3'UTR isoforms, are sufficient to promote mRNA localization to distal compartments of neurons using fluorescent reporters, as a standard method. This would add new biology. On the computational analysis side, the authors should assess if G-quadruplexes are enriched in their dendritic transcriptome datasets, consistent with a recent paper in *eLife* on the role of FMRP interactions with G-quadruplexes in mRNA localization (Goering et al., 2020). This topic is not addressed. The authors previously published a paper that dismissed the roles of 3'UTR and instead revealed that FMRP interacts promiscuously mostly along coding sequences to stall ribosomes and repress translation of autism linked mRNAs (Darnell et al., Cell 2011). The present data sets need to be overlapped and compared to their previous paper to identify relationships or lack thereof. Lastly is an important issue of whether or not there are consensus motifs for FMRP binding as suggested by several recent bioinformatics studies. This topic can be addressed and would add new biology to the present study on how FMRP achieves selectivity.

*Reviewer #1 (Recommendations for the authors):*

Despite the fact that FMRP targets are overrepresented in the dendritic transcriptome, it does not appear from this study that FMRP plays an active role in the mechanism of dendritic mRNA localization, at least under steady state conditions. One goal of the manuscript is to address a major question in the mRNA localization field, which is how FMRP may differentially modulate "localization" of functional classes of mRNAs such as those encoding transcriptional regulators and synaptic plasticity genes (Line 78-90). The data here indicate that FMRP directly interacts with functional classes of mRNAs in different cellular compartments, which has previously been shown in the field. However, no evidence is provided that mechanistically reveal a role for FMRP to promote subcellular localization of different functional classes of mRNAs. The correlative evidence presented in this manner does not add mechanistic insight. The authors can strengthen the manuscript by providing additional experimental evidence and/or computational analysis. Related to this, it is unclear how the authors reconcile the data in Figure 7E with their overarching model that increased levels of FMRP binding actively promote mRNA localization, as suggested by Figure 5 and 6. However, the data in Figure 7E indicate that mRNA localization is not perturbed in the absence of FMRP.

Further related to a role of FMRP in mRNA localization, a recent paper in *eLife* reports that FMRP RGG box promotes mRNA localization of a set of FMRP targets through G-quadruplexes (Goering et al., 2020). This relevant paper needs to be cited and discussed. This relates to earlier work from the Darnell lab that identified a substantial pool of FMRP targets mRNAs having G-quadruplexes (Darnell et al., Cell 2001). The authors should investigate if G-quadruplexes are enriched in their dendritic transcriptome datasets and if not, this might help explain their findings in Figure 7E and 7F.

In Figure 5, there are missing key validation experiments showing that the 3'UTRs of Ankrd11 and Cnksr2 with high levels of FMRP-CLIP reads are sufficient to promote mRNA localization to distal compartments of neurons. The data in their current form are correlative and suggest that mRNAs containing high levels of FMRP are localized to the neuropil, however this is not validated with experimental evidence. Given these experiments are extremely difficult to perform in vivo, a simpler model in cultured neurons with reporter assays would be sufficient and support their claims with functional evidence.

Figure 7 describes critical experiments in Fmr1 KO mice to analyze mRNA dysregulation in cell body and neuropil compartments, integrating with the analysis of differentially localized 3'UTR isoforms, as well alternatively spliced isoforms that were identified earlier in the paper. The analysis and presentation of Figure 7 needs more depth to better tie together the role of FMRP in dendritic mRNA regulation and isoform regulation. Figure 7 needs to show the data in the same format as earlier figures. For example, an analysis of "dendrite present" and "dendrite enriched" mRNAs (Figure 1) needs to be presented and analyzed in Fmr1 KO KO in same manner as shown in earlier figures for WT neurons. Furthermore, the analysis of isoforms in Figures2+3 in WT needs to be compared to KO in Figure 7 – analyzed and presented in the same manner. Lastly, it would be helpful to show quantitative FISH for FMRP targets in WT vs. KO.

In Figure 1-3, the majority of the data is based on combining bulk RNA-sequencing and TRAP data into categories referred to as "dendrite-present" or "dendrite-enriched". By analyzing the data for only ribosome-associate transcripts, the authors eliminate transcripts that are not associated with ribosomes. The authors could use only the bulk-sequencing transcriptomics data, which would be more appropriate for their questions and interpretations. Alternatively, the authors should change the names of the categories to indicate that the transcripts in these categories are specifically ribosome-associated.

Some additional discussion and evaluation of relevant literature is needed to explain what aspects fit or do not fit with the proposed model. FMRP biology is more complex and this study should tie together and integrate different mechanisms on translational control (both negative and positive regulation) and mRNA stability.

– The authors should tie a few key earlier papers of their own research with the present study into a cohesive model, explaining some knowledge gaps on the role of FMRP domains, mRNA binding and precise mechanisms of translational control (Darnell et al., Cell 2001; Darnell et al., Genes Dev 2005; Darnell et al., Cell 2011).

– The authors should discuss and compare other data obtained with TRAP-Seq in Fmr1 KO (Thompson et al., Neuron 2017).

– The authors should discuss how TRAP-Seq analysis compares to ribosome profiling in Fmr1 KO to infer different types of FMRP dependent mechanisms of localization and translational control (Goering et al., 2020; Greenblatt et al., Science 2018) as well as mRNA stability (Shu et al., PNAS 2020).

*Reviewer #2 (Recommendations for the authors):*

– Include a Venn diagram of CA1 FMRP targets overlapping with G-quartet mRNAs.

– Organization of figures and text would improve the manuscript (it was not clear why FMRP CLIP was introduced in the middle and not earlier).

– The abstract should be reworded to better reflect the current version of the manuscript.

---

## [Author Response]

Essential revisions:Two key experiments are needed to support the claim that FMRP plays a role in localization, which is only correlative from the data shown. First is to perform quantitative analysis of FMRP target mRNA localization in dendrites from WT vs. Fmr1 KO mice.

While we have now looked in detail for a role for FMRP in localization, both in response to Reviewer’s suggestions and to some degree in our initial submission, we explicitly do not find evidence for such a role. This conclusion rests on undertaking the suggested experiments, most importantly quantitative RNA analysis (FISH, RNA-seq and TRAP-Seq) of FMRP target localization in dendrites of WT and FMRP KO mouse brain. We now include supplements to Figure 7 (supplements 2 and 3), which shows data from experiments performing FISH on FMRP WT and KO mouse brain slices. We performed global analysis of 3’UTR usage in FMRP WT/KO, similar to Figure 2. We also analyzed FMRP targets whose mRNAs were found to be dendrite-enriched in our studies (and now present this data for Kmt2d, Lrrc7 and Map2 in Figure 7 – supplement 2), and find no significant differences in mRNA localization in FMRP KO mouse brain slices in these studies.

We also searched for but did not find evidence for changes in differential 3’UTR isoform usage in the two compartments in FMRP KO mice. Globally, analysis of all 3’UTRs revealed no significant change in usage between FMRP KO and WT animals (Figure 7 – supplement 3A). We validated these by FISH for two genes (Cnksr2 and Anks1b) which are FMRP targets that produce 3’UTR isoforms that are differentially bound by FMRP and localized to the dendrites (Figure 7 – supplement 3). In sum, do not find any evidence for significant dysregulation of mRNA localization in these genes in FMRP KO mice (Figure 7 – supplement 3).

A second additional key validation experiment is to show that 1 or 2 mRNAs with high levels of FMRP-CLIP reads, for example, specific 3'UTR isoforms, are sufficient to promote mRNA localization to distal compartments of neurons using fluorescent reporters, as a standard method.

Given the results found in Figure 7 – supplements 2 and 3 which indicate that FMRP does not play a role in localization of mRNAs in the CA1 region but rather is involved in regulation of their translation, we do not believe these studies are warranted.

This would add new biology. On the computational analysis side, the authors should assess if G-quadruplexes are enriched in their dendritic transcriptome datasets, consistent with a recent paper in eLife on the role of FMRP interactions with G-quadruplexes in mRNA localization (Goering et al., 2020). This topic is not addressed.

We apologize for this omission, and have performed this analysis and present the results in detail below in the response to Reviewer 1 and in the new Figure 4 – supplement 1 in the manuscript. To summarize, we do not find evidence in our data of G-quadruplexes (either experimentally-defined or G-quadruplex motifs) playing a role in determination of FMRP binding in CA1 dendrites.

The authors previously published a paper that dismissed the roles of 3'UTR and instead revealed that FMRP interacts promiscuously mostly along coding sequences to stall ribosomes and repress translation of autism linked mRNAs (Darnell et al., Cell 2011). The present data sets need to be overlapped and compared to their previous paper to identify relationships or lack thereof. Lastly is an important issue of whether or not there are consensus motifs for FMRP binding as suggested by several recent bioinformatics studies. This topic can be addressed and would add new biology to the present study on how FMRP achieves selectivity.

We appreciate this suggestion and acknowledge that the question of how FMRP recognizes its targets is highly interesting and elusive. However, in this work, consistent with our previous work presenting FMRP-CLIP on both whole brain (J. C. Darnell et al., 2011) and in CA1 neurons (Sawicka et al., 2019), we find FMRP binding patterns that are consistent with FMRP coating the CDS and proximal 3’UTR sequences of its targets. We do not see reliable, reproducible evidence for “peaks” like those present in CLIP experiments on RNA-binding proteins that exhibit sequencing-specific binding patterns. Therefore it is not possible to generate specific sequences where FMRP seems to be binding for input into motif analyses programs, and any motif analysis performed on FMRP CLIP tag sequences themselves would likely reflect the nature of the mRNAs themselves (i.e. GC-content, codon-optimality) and not give information about specificity of FMRP binding.

In addition, our previous cell-type specific FMRP-CLIP experiments have been on bulk neuronal material (Sawicka et al. 2019; Van Driesche et al., n.d.). Although cell-type specific TRAP-seq has been performed on microdissected CA1 compartments (Ainsley et al. 2014), investigators were unable to isolate significant amounts of RNA from resting neurons, and degradation of the isolated RNAs did not allow the types of 3’UTR and alternative splicing analyses that were performed here. The Schuman group has performed extensive analysis of mRNAs from microdissected CA1 compartments (Cajigas et al. 2012a; Tushev et al. 2018), but have not performed FMRP-CLIP or any experiments using cell-type specific or direct protein-RNA regulatory methods. In vitro systems have been used to analyze mRNA localization in FMRP KO systems (i.e. (Goering et al. 2020)), but in vitro systems are unable to fully recapitulate the complexities of in vivo brain regions, and did not analyze direct RNA-protein interactions. As our work is on in vivo brain slices, is cell-type specific, and integrates TRAP-seq, PAPERCLIP and CLIP-seq datasets, we believe that our work is novel and will be of great interest to the field.

Reviewer #1 (Recommendations for the authors):Despite the fact that FMRP targets are overrepresented in the dendritic transcriptome, it does not appear from this study that FMRP plays an active role in the mechanism of dendritic mRNA localization, at least under steady state conditions. One goal of the manuscript is to address a major question in the mRNA localization field, which is how FMRP may differentially modulate “localization” of functional classes of mRNAs such as those encoding transcriptional regulators and synaptic plasticity genes (Line 78-90). The data here indicate that FMRP directly interacts with functional classes of mRNAs in different cellular compartments, which has previously been shown in the field. However, no evidence is provided that mechanistically reveal a role for FMRP to promote subcellular localization of different functional classes of mRNAs. The correlative evidence presented in this manner does not add mechanistic insight. The authors can strengthen the manuscript by providing additional experimental evidence and/or computational analysis. Related to this, it is unclear how the authors reconcile the data in Figure 7E with their overarching model that increased levels of FMRP binding actively promote mRNA localization, as suggested by Figure 5 and 6. However, the data in Figure 7E indicate that mRNA localization is not perturbed in the absence of FMRP.

We appreciate this point, and have provided the suggested analyses. As discussed above, our data does not suggest a role for FMRP in mRNA localization, but rather a preference for FMRP binding to localized mRNA isoforms and a role for FMRP in regulating ribosome association in the dendrites. We detailed above the changes to the manuscript that address this issue.

Nonetheless, we did do additional experimental and computational analysis to strengthen the paper overall and specifically with respect to a role for FMRP in localization. Additional experimental analyses now included in the paper include FISH, and computational analysis includes detailed analysis of the dendritic transcriptome of FMRP KO animals and G-quadruplex studies, as above.

Further related to a role of FMRP in mRNA localization, a recent paper in eLife reports that FMRP RGG box promotes mRNA localization of a set of FMRP targets through G-quadruplexes (Goering et al., 2020). This relevant paper needs to be cited and discussed. This relates to earlier work from the Darnell lab that identified a substantial pool of FMRP targets mRNAs having G-quadruplexes (Darnell et al., Cell 2001). The authors should investigate if G-quadruplexes are enriched in their dendritic transcriptome datasets and if not, this might help explain their findings in Figure 7E and 7F.

We have performed this analysis and discussed the findings above. In addition, we have added Figure 4 – supplement 1, which shows the results of this analysis.

In Figure 5, there are missing key validation experiments showing that the 3'UTRs of Ankrd11 and Cnksr2 with high levels of FMRP-CLIP reads are sufficient to promote mRNA localization to distal compartments of neurons. The data in their current form are correlative and suggest that mRNAs containing high levels of FMRP are localized to the neuropil, however this is not validated with experimental evidence. Given these experiments are extremely difficult to perform in vivo, a simpler model in cultured neurons with reporter assays would be sufficient and support their claims with functional evidence.

We addressed this by performing a global analysis of 3’UTR usage in dendrites and cell bodies of FMRP WT and KO mice and find no evidence for FMRP-dependent localization of 3’UTR isoforms (Figure 7 – supplement 3). We also specifically performed FISH to detect 3’UTR isoforms of Cnksr2 and Anks1b in FMRP KO and WT brain slices to assess whether 3’ UTRs of FMRP targets showed FMRP-dependent changes in localization. In none of these instances did we find any evidence for such changes in dendritic localization in FMRP KO (see Figure 7 – supplement 3).

Figure 7 describes critical experiments in Fmr1 KO mice to analyze mRNA dysregulation in cell body and neuropil compartments, integrating with the analysis of differentially localized 3'UTR isoforms, as well alternatively spliced isoforms that were identified earlier in the paper. The analysis and presentation of Figure 7 needs more depth to better tie together the role of FMRP in dendritic mRNA regulation and isoform regulation. Figure 7 needs to show the data in the same format as earlier figures. For example, an analysis of "dendrite present" and "dendrite enriched" mRNAs (Figure 1) needs to be presented and analyzed in Fmr1 KO KO in same manner as shown in earlier figures for WT neurons. Furthermore, the analysis of isoforms in Figures2+3 in WT needs to be compared to KO in Figure 7 – analyzed and presented in the same manner. Lastly, it would be helpful to show quantitative FISH for FMRP targets in WT vs. KO.

We have included a new Figure 7 – supplement 2 (also included above) which compares dendrite-enriched and dendrite-present mRNAs in FMRP WT and KO neurons and also Figure 7 – supplement 3, which analyzes 3’UTR usage in WT and KO neurons. Importantly, as discussed above, we do not see major differences in the dendrite-enriched or dendrite-present lists when called on FMRP WT vs KO animals, and we don’t see altered distribution of 3’UTR isoforms. This is consistent with our finding that FMRP does not play a role in mRNA localization in resting CA1 neurons.

In Figure 1-3, the majority of the data is based on combining bulk RNA-sequencing and TRAP data into categories referred to as "dendrite-present" or "dendrite-enriched". By analyzing the data for only ribosome-associate transcripts, the authors eliminate transcripts that are not associated with ribosomes. The authors could use only the bulk-sequencing transcriptomics data, which would be more appropriate for their questions and interpretations. Alternatively, the authors should change the names of the categories to indicate that the transcripts in these categories are specifically ribosome-associated.

We appreciate the Reviewer’s concerns, but would note that bulk-sequencing would not work for cell-type specific studies, a key attribute of this study and what distinguishes our work from others, such as the work of Erin Schuman’s group, which has extensively described the bulk transcriptome of the CA1 neuropil (Cajigas et al., 2012a; Tushev et al., 2018).

We acknowledge that TRAP data only describes ribosome-bound mRNAs and therefore cannot capture mRNAs that are completely devoid of bound ribosomes. We address this in several ways. First, we supplement TRAP data with PAPER-CLIP data, which captures all polyadenylated CA1 transcripts, independent of whether they are ribosome-bound. PolyA-bound mRNA data is concordant with TRAP data, suggesting that the two are largely overlapping (Figure 2 – supplement 1). We now discuss this more clearly in the manuscript, and also present results for both bulk RNA-seq and TRAP in Figure 7 as requested by the Reviewer.

Some additional discussion and evaluation of relevant literature is needed to explain what aspects fit or do not fit with the proposed model. FMRP biology is more complex and this study should tie together and integrate different mechanisms on translational control (both negative and positive regulation) and mRNA stability.

We agree that over time it has become evident that FMRP biology is more complex than previously known. Most importantly, we now clearly find that FMRP is binding different mRNAs in the CA1 cell body and dendritic layers, and suggest that FMRP-mediated regulation results in different outcomes in the cell bodies and dendrites of CA1 neurons. We have revised our model (shown above) to incorporate this more detailed model, and have also included more context in the Discussion for how these findings fit into the current knowledge of the molecular biology of FMRP.

– The authors should tie a few key earlier papers of their own research with the present study into a cohesive model, explaining some knowledge gaps on the role of FMRP domains, mRNA binding and precise mechanisms of translational control (Darnell et al., Cell 2001; Darnell et al., Genes Dev 2005; Darnell et al., Cell 2011).

We agree that it remains unknown how FMRP recognizes specific targets, e.g. chromatin-associated or synaptic targets from the larger pool of transcripts in CA1 neurons (and elsewhere). What we have now added to the literature is that it does bind different targets in different compartments and FMRP regulation in these subcellular compartments may lead to different outcomes. In order to better represent our findings, we have revised our model figure to give a more mechanism view of the different modes of FMRP regulation in the two compartments, and have also added an additional section in the Discussion that ties together our previous work, other work in the field and the present work.

– The authors should discuss and compare other data obtained with TRAP-Seq in Fmr1 KO (Thompson et al., Neuron 2017).– The authors should discuss how TRAP-Seq analysis compares to ribosome profiling in Fmr1 KO to infer different types of FMRP dependent mechanisms of localization and translational control (Goering et al., 2020; Greenblatt et al., Science 2018) as well as mRNA stability (Shu et al., PNAS 2020).

We appreciate this suggestion, and agree that our manuscript would benefit from more context between our work and other groups. We address this in two ways. First, we have revised our model figure in order to better detail the different outcomes of FMRP regulation in the two compartments, with the results in the cell bodies being similar to those found for whole-cell experiments in other works (i.e. Thomson, Coelin, Shu). No subcellular compartment-specific CLIP or ribosome profiling in FMRP WT or KO animals has been performed, making our work novel and of great importance to the field. Further, our results in dendrites globally- and FMRP-target specifically- confirm the hypothesis of excess local translation in the synapses of FMRP KO animals.

Reviewer #2 (Recommendations for the authors):– Include a Venn diagram of CA1 FMRP targets overlapping with G-quartet mRNAs.

As discussed above in response to a similar suggestion by Reviewer 1, we have performed enrichment analysis to determine whether G-quadruplex containing mRNAs are enriched in dendrite-enriched FMRP target mRNAs when compared to dendrite-enriched FMRP non-targets. We do not find evidence of significant enrichment, but have included the results of this analysis in a new supplement to Figure 4 (supplement 1) of the manuscript, which we included and discussed above.

– Organization of figures and text would improve the manuscript (it was not clear why FMRP CLIP was introduced in the middle and not earlier).

Thank you for this suggestion and have made an effort to increase the readability of the manuscript. We wish to point out the novelty of the system established and analyses performed in Figures 1-3, before FMRP was introduced. We believe that our use of TRAP-seq and PAPERCLIP in order to define the cell-type- and compartment-specific transcriptome of CA1 neurons is novel and of great value to the field in and of itself, and only in this analysis were we to find that FMRP targets made up a significant fraction of dendritic mRNAs, which we then followed up by performing compartment-specific FMRP-CLIP experiments.

– The abstract should be reworded to better reflect the current version of the manuscript.

We appreciate this suggestion and have done so.

References

Ainsley, Joshua A., Laurel Drane, Jonathan Jacobs, Kara A. Kittelberger, and Leon G. Reijmers. 2014. “Functionally Diverse Dendritic mRNAs Rapidly Associate with Ribosomes Following a Novel Experience.” *Nature Communications* 5 (July): 4510.

Bear, Mark F., Kimberly M. Huber, and Stephen T. Warren. 2004. “The mGluR Theory of Fragile X Mental Retardation.” *Trends in Neurosciences* 27 (7): 370–77.

Biswas, Jeetayu, Leti Nunez, Sulagna Das, Young J. Yoon, Carolina Eliscovich, and Robert H. Singer. 2019. “Zipcode Binding Protein 1 (ZBP1; IGF2BP1): A Model for Sequence-Specific RNA Regulation.” *Cold Spring Harbor Symposia on Quantitative Biology* 84: 1–10.

Cajigas, Iván J., Georgi Tushev, Tristan J. Will, Susanne tom Dieck, Nicole Fuerst, and Erin M. Schuman. 2012a. “The Local Transcriptome in the Synaptic Neuropil Revealed by Deep Sequencing and High-Resolution Imaging.” *Neuron* 74 (3): 453–66.

Cajigas, Iván J., Georgi Tushev, Tristan J. Will, Susanne tom Dieck, Nicole Fuerst, and Erin M. Schuman. 2012b. “The Local Transcriptome in the Synaptic Neuropil Revealed by Deep Sequencing and High-Resolution Imaging.” *Neuron* 74 (3): 453–66.

Darnell, Jennifer C., Sarah J. Van Driesche, Chaolin Zhang, Ka Ying Sharon Hung, Aldo Mele, Claire E. Fraser, Elizabeth F. Stone, et al. 2011. “FMRP Stalls Ribosomal Translocation on mRNAs Linked to Synaptic Function and Autism.” *Cell* 146 (2): 247–61.

Darnell, Robert B. 2020. “The Genetic Control of Stoichiometry Underlying Autism.” *Annual Review of Neuroscience* 43 (July): 509–33.

Eom, Taesun, Chaolin Zhang, Huidong Wang, Kenneth Lay, John Fak, Jeffrey L. Noebels, and Robert B. Darnell. 2013. “NOVA-Dependent Regulation of Cryptic NMD Exons Controls Synaptic Protein Levels after Seizure.” *eLife* 2 (January): e00178.

Goering, Raeann, Laura I. Hudish, Bryan B. Guzman, Nisha Raj, Gary J. Bassell, Holger A. Russ, Daniel Dominguez, and J. Matthew Taliaferro. 2020. “FMRP Promotes RNA Localization to Neuronal Projections through Interactions between Its RGG Domain and G-Quadruplex RNA Sequences.” *eLife* 9 (June). https://doi.org/10.7554/*eLife*.52621.

Guo, Junjie U., and David P. Bartel. 2016. “RNA G-Quadruplexes Are Globally Unfolded in Eukaryotic Cells and Depleted in Bacteria.” *Science* 353 (6306). https://doi.org/10.1126/science.aaf5371.

Lee, Hye Young, Woo-Ping Ge, Wendy Huang, Ye He, Gordon X. Wang, Ashley Rowson-Baldwin, Stephen J. Smith, Yuh Nung Jan, and Lily Yeh Jan. 2011. “Bidirectional Regulation of Dendritic Voltage-Gated Potassium Channels by the Fragile X Mental Retardation Protein.” *Neuron* 72 (4): 630–42.

Sawicka, Kirsty, Caryn R. Hale, Christopher Y. Park, John J. Fak, Jodi E. Gresack, Sarah J. Van Driesche, Jin Joo Kang, Jennifer C. Darnell, and Robert B. Darnell. 2019. “FMRP Has a Cell-Type-Specific Role in CA1 Pyramidal Neurons to Regulate Autism-Related Transcripts and Circadian Memory.” *eLife* 8 (December). https://doi.org/10.7554/*eLife*.46919.

Tushev, Georgi, Caspar Glock, Maximilian Heumüller, Anne Biever, Marko Jovanovic, and Erin M. Schuman. 2018. “Alternative 3′ UTRs Modify the Localization, Regulatory Potential, Stability, and Plasticity of mRNAs in Neuronal Compartments.” *Neuron*. https://doi.org/10.1016/j.neuron.2018.03.030.

Van Driesche, Sarah J., Kirsty Sawicka, Chaolin Zhang, Sharon K. Y. Hung, Christopher Y. Park, John J. Fak, Chingwen Yang, Robert B. Darnell, and Jennifer C. Darnell. n.d. “FMRP Binding to a Ranked Subset of Long Genes Is Revealed by Coupled CLIP and TRAP in Specific Neuronal Cell Types.” https://doi.org/10.1101/762500.

Zappulo, Alessandra, David van den Bruck, Camilla Ciolli Mattioli, Vedran Franke, Koshi Imami, Erik McShane, Mireia Moreno-Estelles, et al. 2017. “RNA Localization Is a Key Determinant of Neurite-Enriched Proteome.” *Nature Communications* 8 (1): 583.